# RAT-Bench: benchmarking Re-identification risk in Anonymized Text

## Abstract

Data containing sensitive personal information is increasingly used to train, fine-tune, or query Large Language Models (LLMs), raising the risk such data may be inadvertently leaked. Text is typically scrubbed of identifying information prior to use, often with tools such as Microsoft's Presidio or Anthropic's PII purifier. These tools are generally evaluated based on their ability to remove manually annotated identifiers (e.g., names), yet their effectiveness at preventing re-identification remains unclear. We introduce RAT-Bench, which is to the best of our knowledge, the first modern, synthetic benchmark for measuring the effectiveness of text anonymization tools at preventing re-identification. We use U.S. demographic statistics to generate synthetic, yet realistic text, that contains various direct and indirect identifiers across diverse domains and levels of difficulty. We apply a range of NER- and LLM-based text anonymization tools on our benchmark and, based on the attributes an LLM-based attacker is able to infer correctly from the anonymized text, we report the risk that any individual will be correctly re-identified in the U.S. population. Our results show that existing tools still often miss direct identifiers or leave enough indirect information for successful re-identification. Indeed, even for the widely used anonymizer Azure and state-of-the-art GPT-4 anonymizer instantiated with the Rescriber prompt, the success rate remains at 29% and 27%, respectively. We conduct ablations for number and type of attribute, and also study the utility and cost of anonymization. We find that NER-based methods can reduce re-identification risk substantially, albeit sometimes at a strong cost in utility. LLM-based tools remove identifiable information more precisely, yet require a higher computational cost. We will release the benchmark and encourage community efforts to expand it, so it remains a robust test as tools become better in the future.

## 1 Introduction

Domain-specific text is essential for advancing today's AI models and systems. Modern Large Language Models (LLMs) (OpenAI, 2025a; Grattafiori et al., 2024) are not only trained on vast amounts of publicly available data (Brown et al., 2020; Touvron et al., 2023) but are also frequently fine-tuned on specialized datasets to perform specific tasks. For example, AI companies, including OpenAI (OpenAI, 2025b) and Anthropic (TechCrunch, 2025) leverage user–chatbot interactions to improve their models (King et al., 2025), a practice recognized as key to generating a sustainable competitive advantage (Huang & Grady, 2023). Moreover, specialized models have been developed for high-stakes domains such as medicine (Kraljevic et al., 2022) and law (FinancialTimes, 2025).

LLMs have, however, been shown to memorize portions of their training data and be susceptible to membership inference attacks (Meeus et al., 2024; Shi et al., 2024; Hayes et al., 2025), or even reproduce training sequences verbatim (Carlini et al., 2021; Nasr et al., 2025; Cooper et al., 2025). Several mitigation strategies have been proposed, but these remain limited in effectiveness or practicality. Deduplicating training data can reduce memorization (Kandpal et al., 2022; Lee et al., 2022), yet models are still capable of memorizing across quite dissimilar sequences (Shilov et al., 2024) and training with formal privacy guarantees requires specialized expertise and often incurs significant utility costs (McKenna et al., 2025). As a result, when domain-specific text is incorporated into training, it is currently impossible to guarantee that sensitive sequences, including personal or confidential information, will not be memorized and, possibly, leaked.

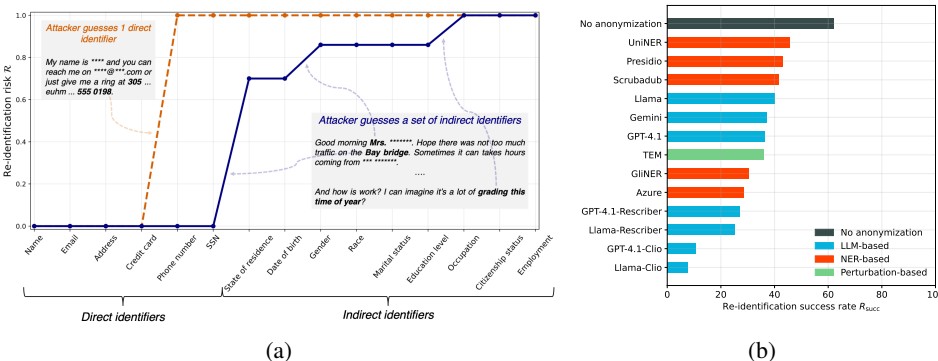

(a)                                                                  (b)

Figure 1: **Evaluating text anonymization through re-identification risk.** (a) We compute re-identification risk $\mathcal{R}$ for two pieces of anonymized text as an attacker aims to sequentially infer any direct or indirect identifiers. When at least one direct identifier (e.g. phone number, illustrated in orange) is inferred, the risk is set to 1. If not, the risk is computed (see Alg 2) based on the set of indirect identifiers (e.g. state, gender and occupation, illustrated in blue) which, together, uniquely identify an individual. The numbers and examples in this graph are provided for illustration only. (b) Re-identification success rate $R_{\text{succ}}$ by anonymization tool, averaged across difficulty levels and scenarios (Table 1). Higher $R_{\text{succ}}$ means a larger fraction of individuals are correctly re-identified from the anonymized text (worse privacy); lower indicates better anonymization.

As a solution, a range of methods have been proposed to *anonymize* text before providing it to an AI model. Approaches range from named entity recognition (NER) combined with heuristics (Microsoft, 2021; Azure, 2023; LeapBeyond, 2023) to recent methods that leverage LLMs for anonymization (Altalla' et al., 2025; Tamkin et al., 2024b; Staab et al., 2025). While such techniques primarily aim at removing *direct identifiers* like names and email addresses, their effectiveness at mitigating the broader risk of re-identification remains unclear (Singh & Narayanan, 2025; Pilán et al., 2022). If insufficiently mitigated, re-identification can expose sensitive information, undermine user trust, and introduce security risks. Moreover, privacy laws reinforce this by requiring data to be considered 'anonymous' (EU's GDPR (2016)) or 'de-identified' (U.S. CCPA (2018)) only when the risk of re-identification, directly or indirectly, is reasonably low. This imposes a higher standard, demanding the consideration not only of direct identifiers but also of *indirect identifiers* that are not unique to an individual in isolation but may be when combined (e.g. zip code, year of birth, or occupation). A large body of work has found seemingly innocuous attributes to be highly identifying, e.g. ranging from anonymized taxi trajectories (Douriez et al., 2016) to mobile phone data (De Montjoye et al., 2013). Notably, Rocher et al. (2019) recently found that as few as 15 demographic attributes can uniquely single out 99.98% of Americans.

**Contribution.** We introduce RAT-Bench, a benchmark to evaluate text anonymization tools through re-identification risk, considering an attacker's inference of both direct and indirect identifiers.

We first generate synthetic, yet realistic, texts grounded in real-world demographics by sampling indirect identifiers (e.g. date of birth, race) from the 5% Public Use Microdata Sample (PUMS) (US-CensusBureau) and augmenting them with consistent direct identifiers (e.g. name, email). From these, we generate benchmark entries in which the identifiers are deliberately mentioned.

Next, we apply 13 anonymization tools from the literature, either NER-, perturbation-, or LLM-based to the benchmark, and instantiate a state-of-the-art attacker (Patsakis & Lykousas, 2023; Staab et al., 2024) which attempts to recover attributes from the anonymized text. If the attacker infers at least one direct identifier, an individual is re-identified (re-identification risk $\mathcal{R} = 1$) and, if not, we use the framework from Rocher et al. (2019) to compute the risk that the individual is correctly re-identified in the entire U.S. population based on inferred indirect identifiers (Figure 1a).

We then compare the risk of re-identification that remains after anonymization. Figure 1b shows how all tools reduce the risk from the non-anonymized baseline, from on average 62% to 32%. Importantly, even for the widely used anonymizer Azure and for an anonymizer based on the highly capable GPT-4.1 instantiated with the Rescriber prompt, the success rate remains at 29% and 27%,

respectively, indicating that substantial identifying information can persist in anonymized texts. We further disentangle this between direct and indirect identifiers, across different levels of difficulty (Figure 2). Even if methods mask out all direct identifiers, our benchmark shows that indirect identifiers meaningfully contribute to re-identification. Ablations of the number of identifiers confirm these trends. We also evaluate a stronger, iterative LLM-based anonymizer (Staab et al., 2025), and find this to substantially reduce the risk, while also highlighting the difficulty of specifying its attributes for new settings.

Lastly, we compare the re-identification rate after anonymization to the cost in utility and computation for each anonymization tool. We find that NER-based methods may remove text overly aggressively, resulting in reduced utility (e.g., BLEU score of $0.57$ for Azure). LLM-based anonymizers offer significantly better privacy-utility tradeoff, by more precisely removing the information necessary for re-identification – albeit at a larger computational cost, especially when applied at scale.

## 2 BACKGROUND AND RELATED WORK

**Text anonymization methods.** Given a text $t$, an anonymization tool $\mathcal{T}$ aims to remove personal, sensitive and/or identifiable information, producing anonymized text $t^a = \mathcal{T}(t)$.

Named Entity Recognition (NER) has long been the dominant approach for removing sensitive information from text (Deußer et al., 2025b). NER models are trained to identify spans corresponding to categories such as names or locations. Detected entities are then masked (with e.g. '**'), yielding an anonymized version of the text. Numerous NER models, such as Flair (Akbik et al., 2019) or Spacy (Honnibal et al., 2020), can be used as such to remove both direct and indirect identifiers, while many tools extend NER with rule-based heuristics and regular-expression matching (Microsoft, 2021; Azure, 2023; Kleinberg et al., 2022; LeapBeyond, 2023). While NER-based systems detect clear identifiers such as emails, they might not have been trained for domain-specific contexts (Singh & Narayanan, 2025). To address this, specialized scrubbers have been developed for clinical (Johnson et al., 2020; Vakili et al., 2022) and legal data (Oksanen et al., 2022).

Further work explored adding controlled, word-level perturbations protect the privacy of a given text. For each word, Madlib (Feyisetan et al., 2020) adds noise in an embedding space before projecting back to the nearest word, while TEM (Carvalho et al., 2023) improves utility by sampling replacements from a distance-weighted distribution over candidate words.

Recent work has also applied LLMs to anonymization. Anthropic (2024) provides *PII purifier*, a general prompt for LLMs to mask identifying information in a piece of text. Zhou et al. (2025) develop a more specific prompt for smaller language models such as LLaMA-3.1-8B by listing the specific entities that are considered sensitive or personal. Altalla' et al. (2025) shows GPT-3.5/4 can be used to de-identify clinical notes, while Liu et al. (2023) prompt GPT-4 with HIPAA guidelines for zero-shot medical text de-identification. Dou et al. (2024) fine-tunes an LLM to replace self-disclosures of PII, and Deußer et al. (2025a) use LLMs to annotate PII before distilling into smaller NER models. Other approaches use LLMs to also go beyond masking. Tamkin et al. (2024b) develop Clio, a privacy-preserving text summarization tool that relies on Claude. Staab et al. (2025) iteratively reword text to remove identifiers and defined contextual clues, while Yang et al. (2024) add another LLM to better balance privacy and utility. Utpala et al. (2023) prompts on LLM to paraphrase a given piece of text, while sampling from the LLM at a specified temperature. Lastly, Frikha et al. (2024) propose replacing sensitive attributes with plausible alternatives to mislead adversaries.

**Anonymization from a privacy law perspective.** Most privacy regimes consider anonymous data to be out-of-scope of privacy laws (privacy laws do not apply to them, including when transferred internationally). This includes EU's GDPR (2016), California's CCPA (2018), and others such as Brazil'sLGPD (2018) or Singapore's PDPA (2012). While legal terminologies (anonymous in the EU vs de-identified in the US) and specific definitions vary, most consider data anonymous if an individual cannot be identified anymore in it. GDPR Rec 26, for instance, uses the "reasonably likely" standard that a natural person cannot be identified "directly or indirectly". Technically this has long been translated to the removal of direct identifiers such as a phone number, email address, or social security number and a low risk of re-identification from indirect identifiers, such as gender, ethnicity, or hometown, where low is defined from the context. UK's standard for health data, e.g., is a K of 5 (UK ICO, 2025), which translates to a correctness value strictly lower than 0.2. Inter-

---

**Algorithm 1:** Benchmark Construction for Text Anonymization Evaluation

---

**Input:** Dataset $D$, scenario $s$, target attributes $\mathcal{A}$, difficulty level $\ell \in \{1, 2, 3\}$, length $N$
**Output:** Benchmark $\mathcal{B}$ consisting of $N$ $(x, t)$ pairs
**for** $i = 1$ **to** $N$ **do**

    record $r \sim D$ ;           // r contains only indirect identifiers $Q(r)$
    $I(r) \leftarrow \text{DIRECTIDGEN}(Q(r))$ ; // Generate direct identifiers (Alg.3)
    $x_{\text{full}} \leftarrow (Q(r), I(r))$ ;               // Concatenate full profile
    $x \leftarrow x_{\text{full}}[\mathcal{A}]$ ;                // Select target attributes
    $P \leftarrow \text{BUILDPROMPT}(x, \ell, s)$ ;       // Construct prompt (Alg.4)
    $t \leftarrow \text{LLM}_{\text{gen}}(P)$ ;                 // Generate text
    $\mathcal{B} \leftarrow \mathcal{B} \cup \{(x, t)\}$ ;             // Add to benchmark

**return** $\mathcal{B}$

---

estingly, UK's recent guidance for the motivated intruder test specifically requires considering the risk of re-identification posed by LLMs (UK ICO, 2025). Some legal regimes, such as EU's GDPR, also define data where direct identifiers have been removed as pseudonymous data. Pseudonymous data, however, remains within the scope of privacy laws (privacy laws apply to them). Finally, PII (Personally Identifiable Information) removal is used as a term by some of the tools, notably Presidio Microsoft (2021) and Azure Azure (2023). While GDPR or CCPA do not use the term, the US Office of Management and Budget, part of the Executive Office of the President, defines PII as data that "distinguish or trace an individual's identity, either alone or when combined with other personal or identifying information that is linked or linkable to a specific individual" and mandates, similarly to privacy laws, a case-by-case assessment of the risk of re-identification taking into account "other available information, [that] could be used to identify an individual" (U.S. GSA, 2025).

**Evaluating text anonymization.** Text anonymization is typically evaluated using recall, i.e. the proportion of manually annotated terms correctly detected (Manzanares-Salor et al., 2024). However, such annotations require a unique ground truth, while private information is often *context dependent, not identifiable, and not discrete* (Brown et al., 2022). To broaden coverage, Pilán et al. (2022) expand manually annotated spans in specific court cases, while Manzanares-Salor et al. (2024) train a classifier to link anonymized documents to those available as background knowledge, thereby addressing disclosure risk. In this work, we quantify how much identifying information an adversary can infer after anonymization and tie this to re-identification risk in the U.S. population.

It has also been shown that LLMs can infer personal attributes from text (Patsakis & Lykousas, 2023; Staab et al., 2024; Liu et al., 2025), a capability used to evaluate anonymization methods (Staab et al., 2024; Yukhymenko et al., 2024). Beyond human-annotated datasets, synthetic data has been used for evaluation: Yukhymenko et al. (2024) generate Reddit-style text seeded with synthetic profiles, while Singh & Narayanan (2025) augment NER benchmarks with LLM-generated variations.

**Alternative solutions.** To enable corpus-level text processing with privacy guarantees, prior work has explored generating synthetic data under differential privacy (Mattern et al., 2022a; Yue et al., 2023; Kurakin et al., 2023). While this has shown promising results for simple settings (Meeus et al., 2025), it comes with an intricate privacy-utility trade-off (McKenna et al., 2025) and requires access to a large set of documents. In contrast, we here focus on anonymizing individual text samples.

## 3 BENCHMARKING TEXT ANONYMIZATION

In this work, we present an end-to-end benchmark for evaluating Reidentification risk in Anonymized Texts (RAT-Bench). We first construct a diverse synthetic dataset grounded in real-world demographics, with controlled occurrences of direct and indirect identifiers. We apply a set of anonmyization tools on our benchmark, and deploy a state-of-the-art LLM attacker to infer identifiers from the anonymized texts. From the inferred identifiers, we obtain an accurate estimate of the re-identification risk across the entire US population. In doing so, our benchmark quantifies re-identification risk using state-of-the-art re-identification techniques, as key to many privacy regulations (GDPR, 2016; CCPA, 2018).

---

**Algorithm 2:** Evaluation of reidentification risk of anonymization tool

---

**Input:** Profile and text $(x, t)$, anonymization tool $\mathcal{T}$, attacker $\text{LLM}_{\text{att}}$, target attributes $\mathcal{A}$:
**Output:** Re-identification risk $\mathcal{R}(x, t, \mathcal{T})$
$t^a \leftarrow \mathcal{T}(t)$ ;                                                    `// Anonymize text`
$\hat{x} \leftarrow \text{LLM}_{\text{att}}(t^a, \mathcal{A})$ ;          `// Infer identifiers from anonymized text`
$x^*_{direct}, x^*_{indirect} \leftarrow extract\_matches(\hat{x}, x)$ ;       `// Compare inference with GT`
**if** $x^*_{direct} \neq \emptyset$ **then**
$\quad \lfloor \quad \mathcal{R} = 1$ ;                                `// Able to infer direct identifier`
**else**
$\quad \lfloor \quad \mathcal{R} = \kappa_x$ ;                              `// Compute re-identification risk`
**return** $\mathcal{R}$

---

**A synthetic benchmark grounded in real-world demographics.** We generate synthetic data grounded in real-world demographics, with the overall procedure summarized in Algorithm 1.

Let $D$ denote a tabular dataset containing indirect identifiers $Q(r)$ for each record $r$. To construct a benchmark of size $N$, we repeatedly sample $r \in D$ and generate text $t$ conditioned on the demographics of $r$. As a first step, since public demographic datasets typically exclude direct identifiers such as names or emails, we produce synthetic ones linked to the chosen indirect identifiers $Q(r)$. For this, a language model $\text{LLM}_{\text{dir}}$ generates realistic but fictitious identifiers $I(r)$, as detailed in Algorithm 3. The indirect and synthetic direct identifiers are then combined into a complete profile $x_{\text{full}}$. A subset of target attributes is then selected as $x = x_{\text{full}}[\mathcal{A}]$, where $A \subseteq Q(r) \cup I(r)$.

Next, we query language model $\text{LLM}_{\text{gen}}$ with a prompt $P = \text{BUILDPROMPT}(x_t, \ell, s)$ (Algorithm 4) to generate text $t$. The prompt is constructed such that the generated text $t$ (i) is situated within a specified scenario $s$ (e.g., a patient–doctor transcript); (ii) conveys information about the target attributes $\mathcal{A}$ of record $r$; and (iii) does so at a specified difficulty level $\ell$. This design allows us to systematically vary scenarios, attributes and difficulty while retaining control over the ground-truth.

We define three difficulty levels: attributes may be (1) stated explicitly in a clean, standard form (easy), (2) explicitly present but in a non-standard form (e.g., slang, nonstandard formatting, partial masking) (medium), or (3) only implied through context or indirect cues, never directly stated (hard). Illustrative examples are shown in Appendix C. For direct identifiers, we restrict to levels 1 and 2, as values such as phone numbers are not realistically implied through contextual cues.

**Evaluation of anonymization tools.** Our evaluation pipeline consists of three stages: anonymization, attack and re-identification risk evaluation. Given a benchmark entry $(x, t)$, where $x = (x_1, \cdots, x_{|\mathcal{A}|})$ is a vector of $|\mathcal{A}|$ direct and indirect identifiers and $t$ is text derived from $x$, we first pass $t$ through the anonymization tool $\mathcal{T}$, resulting in the anonymized text $t^a = \mathcal{T}(t)$.

Next, we instantiate the attacker $\text{LLM}_{\text{att}}$ from Staab et al. (2024) to infer identifiers from the anonymized text. For each attribute $x_i$, the attacker produces a guess $\hat{x}_i = \text{LLM}_{\text{att}}(t^a, A_i)_i$. We compare these guesses to the ground truth and extract the correct guesses $x^* = (x^*_{direct}, x^*_{indirect})$, where $x^*_{direct}$ is the vector of correctly guessed direct identifiers, and analogously for $x^*_{indirect}$.

We provide an estimate of the risk that the individual $r$ associated with anonymized text $t^a$ can still be re-identified by the attacker. If at least one direct identifier is correctly recovered, i.e., $x^*_{\text{direct}} \neq \emptyset$, we deem the profile re-identified and set the re-identification risk to $\mathcal{R} = 1.0$. For example, if an attacker can still successfully infer an individual's phone number from $t^a$, re-identification is successful (see Figure 1a). Otherwise, when $x^*_{\text{direct}} = \emptyset$, we instead compute the risk of re-identification based on the set of correctly guessed indirect identifiers $x^*_{\text{indirect}}$ following the framework from Rocher et al. (2019). Specifically, we compute the probability that the individual corresponding to record $r$ can be correctly identified from the US population using the values of $x^*_{\text{indirect}}$ (*correctness* in Rocher et al. (2019)), or $\mathcal{R} = \kappa_x$. We provide the full evaluation in Alg. 2.

## 4 EXPERIMENTAL SETUP

**RAT-Bench construction.** As tabular dataset $D$ with real-world demographics, we use indirect identifiers collected during the American Community Survey and made available as the 5% Public Use Microdata Sample (PUMS) (USCensusBureau). We select a subset of 9 PUMS attributes also considered by Rocher et al. (2019) as indirect identifiers $Q(r)$: state of residence, gender, date of birth, race, marital status, highest level of education obtained, employment status, occupation and citizenship status. We generate the following 6 direct identifiers $I(r)$: name, social security number (SSN), credit card number, phone number, address, and email address. We provide further details in Appendix A and provide example benchmark entries in Appendix D.

We consider two scenarios $s$: (1) medical appointment transcripts and (2) AI chatbot interactions. These represent realistic targets for anonymization (King et al., 2025; OpenAI, 2025b; Liu et al., 2023; Johnson et al., 2020) and together capture a diverse range of contexts in which attributes may appear. We provide a detailed description of each scenario in Table 3. We then use Algorithm 4 to construct a specialized prompt $P$ for each scenario and difficulty in our benchmark generation.

As target attributes $\mathcal{A}$, we consider $N_i$ direct identifiers and $N_q$ indirect identifiers. For each benchmark entry (i.e. run $i$ out of $N$ in Algorithm 1), we randomly sample $N_i$ attributes from $I(r)$ and $N_q$ from $Q(r)$ to ensure that attributes are evenly distributed across entries and that identifiers co-occur without inducing any dependency. We use Gemini 2.5 Flash (Comanici et al., 2025) as $\text{LLM}_{\text{gen}}$ to generate benchmark $\mathcal{B}$. Unless stated otherwise, we sample 100 records $r$ for each reported aggregation (e.g. values in Table 1) and limit the length of the text to 500 words for difficulty 1-2, and 1000 for level 3. We also use GPT-5 (OpenAI, 2025a) to generate an additional small scale benchmark for comparison. Further details about the comparison of both generators can be found in Appendix B.

**Anonymization tools.** We then use RAT-Bench to evaluate the effectiveness of existing anonymization tools $\mathcal{T}$, broadly distinguishing between tools relying on NER, perturbations and LLMs. *(i) NER-based* approaches rely on NLP models (e.g., BERT (Devlin et al., 2019)) and heuristics such as regular expression matching to detect patterns corresponding to predefined attributes. In this category, we evaluate Azure Language Studio (Azure, 2023), Presidio (Microsoft, 2021), Scrubadub (LeapBeyond, 2023), GliNER (Zaratiana et al., 2024), and UniNER (Zhou et al., 2024). *(ii) Pert.* For perturbation-based methods, we evaluate TEM (Carvalho et al., 2023) with $\epsilon = 11$ (results for Madlib (Feyisetan et al., 2020) and other values of $\epsilon$ in Appendix N). *(iii) LLM-based* approaches use one- or few-shot prompting for anonymization, without task-specific fine-tuning. We compiled three prompts for our experiment: the PII purifier prompt as proposed by Anthropic (Anthropic, 2024), the Clio summarization prompt by Tamkin et al. (2024a) and the Rescriber prompt by Zhou et al. (2025). We use these prompts to initialize GPT-4 (OpenAI, 2023), Gemini 2.5 Flash (Comanici et al., 2025) and Llama 3.1–8B AI@Meta (2024) as LLM-based anonymizers. The exact prompts are presented in Appendix G, alongside ablation results when slight adjustments are made to the Anthropic prompt. We also evaluate DP-Prompt (Utpala et al., 2023) in Appendix N.

**Evaluation metrics.** We instantiate a state-of-the-art attacker to infer attributes from anonymized text $t^a$; we use LLaMA-3.1-8B-Instruct (AI@Meta, 2024) as $\text{LLM}_{\text{att}}$, prompted with the attacker template from Staab et al. (2024). textcolorblueWe also evaluate GPT-4 (OpenAI, 2023) as an attacker model in Appendix L. For each target attribute $a \in \mathcal{A}$ present in the original text $t$, the attacker generates a guess, which is compared against the ground truth for $x^*_{direct}$ and $x^*_{indirect}$ (details see Appendix F). Given the matches, we compute the re-identification risk of record $x$ as $\mathcal{R}(x) = 1$ if $x^*_{direct} \neq 0$, and $\mathcal{R}(x) = \kappa_x$ otherwise. For any subset $S \subseteq \mathcal{B}$ of the benchmark (e.g., a specific scenario $s$ and difficulty level $l$), we report the fraction of successfully re-identified records in the benchmark subset, or the re-identification success rate $R_{\text{succ}}(S) = \frac{1}{|S|} \sum_{x \in S} \mathbf{1}\{\mathcal{R}(x) = 1\}$.

## 5 RESULTS

We first consider benchmark entries with $N_i = 1$ direct and $N_q = 5$ indirect identifiers. Table 1 shows the re-identification success rate $R_{\text{succ}}$ across anonymizers, scenarios, and difficulty levels. For difficulty levels 1 (easy) and 2 (medium), Figure 2 further disentangles the success rate between individuals identified purely based on the 1 direct identifier, and, if this was unsuccessful, individuals re-identified based on the 5 indirect identifiers. Note that for difficulty level $l = 3$ (hard), we do not include any direct identifier and just consider indirect ones.

| Anonymization tool $\mathcal{T}$ | | Medical Conversation | | | | AI Chatbot | | | |
| --- | --- | --- | --- | --- | --- | --- | --- | --- | --- |
| Class | Implementation | Easy | Med. | Hard | **Avg.** | Easy | Med. | Hard | **Avg.** |
| | No anonymization | 88% | 77% | 34% | **66%** | 80% | 66% | 28% | **58%** |
| NER | Azure | 31% | 32% | 29% | **31%** | 37% | 27% | 15% | **26%** |
| | Presidio | 52% | 43% | 29% | **41%** | 58% | 56% | 21% | **45%** |
| | Scrubadub | 48% | 33% | 31% | **38%** | 56% | 57% | 22% | **45%** |
| | Gliner | 40% | 26% | 31% | **32%** | 24% | 43% | 19% | **29%** |
| | Uniner | 58% | 59% | 24% | **47%** | 59% | 51% | 22% | **44%** |
| Pert. | TEM | 45% | 34% | 25% | **35%** | 51% | 35% | 24% | **37%** |
| LLM | *Prompt* — *Model* | | | | | | | | |
| | Anthropic — GPT-4.1 | 47% | 33% | 33% | **37%** | 53% | 34% | 22% | **36%** |
| | Anthropic — Gemini | 52% | 37% | 34% | **41%** | 53% | 47% | 18% | **39%** |
| | Anthropic — Llama | 48% | 39% | 34% | **40%** | 45% | 39% | 18% | **34%** |
| | Clio — GPT-4.1 | 8% | 7% | 10% | **8%** | 15% | 14% | 11% | **13%** |
| | Clio — Llama | 3% | 7% | 4% | **4%** | 16% | 12% | 6% | **11%** |
| | Rescriber — GPT-4.1 | 15% | 36% | 35% | **29%** | 25% | 30% | 22% | **25%** |
| | Rescriber — Llama | 16% | 27% | 28% | **23%** | 27% | 32% | 23% | **27%** |
| **Avg. (across tools)** | | 36% | 31% | 27% | **31%** | 40% | 37% | 19% | **32%** |

Table 1: RAT-Bench re-identification success rate (%) across anonymizers, scenarios, and difficulty.

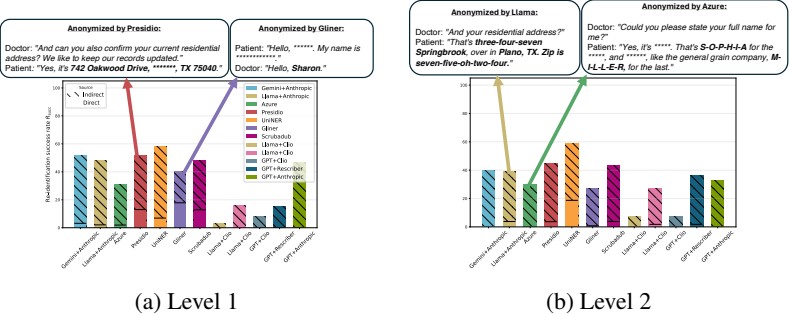

(a) Level 1                    (b) Level 2

Figure 2: Percentage of correctly re-identified individuals for Medical Conversation (Table 1), disentangling re-identification based on (i) direct (unshaded) and (ii) indirect identifiers (shaded). We provide example failure cases for direct identifiers (more in Appendix J.1).

As a baseline, we evaluate re-identification success on the **original, non-anonymized text**. This reflects the attacker's maximum possible success rate at each difficulty level. In both scenarios, the attacker successfully re-identifies profiles with easy and medium attributes in over 65% of cases. This confirms that RAT-Bench entries contain substantial identifying information that is extracted by the attacker. Explicitly mentioned identifiers (easy) are reliably recovered, and the attacker is generally unaffected by the variations at medium difficulty. Even at the hard level, with only indirect identifiers, success rates reach up to 34%, demonstrating that despite the increased difficulty, the attacker is still able to correctly guess a substantial fraction of them.

We then analyze how the attacker's re-identification risk decreases on the **anonymized text**, using a range of tools from the literature. For all tools, the average risk decreases: for Medical Conversations from 66% to 31%, and for AI Chatbot from 58% to 32%. Llama (Clio) performs the best, reducing the average re-identification risk to 4% (Medical Conversation) and 11% (AI Chatbot). However, Clio prompts the LLM to summarize the entire text into at most two sentences, resulting in significant utility loss. Other than Clio, Llama (Rescriber) perform the best, reducing the risk on average to 23% (Medical Conversations) and 27% (AI Chatbot). Among the non-LLM based tools, Azure performs the best, reducing the risk on average to 31% (Medical Conversations) and 26% (AI Chatbot). Importantly, average success rates for all tools (excluding LLMs using Clio prompts) remain above 30%, indicating that substantial identifying information often persists.

To further understand why the success rate remains as high after anonymization, we examine the risk **per difficulty level**. Across scenarios and tools, the largest risk reduction occurs at the **easy level**,

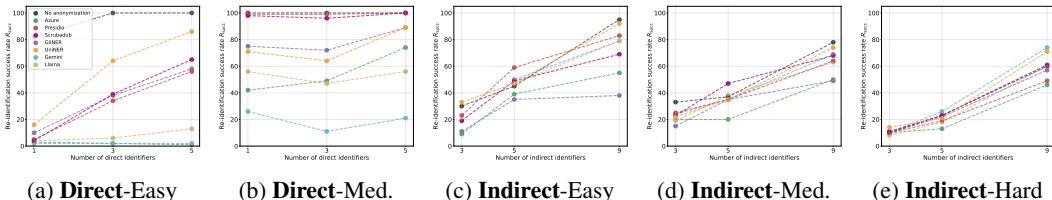

Figure 3: Re-identification success rate for increasing number of direct (a-b) and indirect identifiers (c-e). Results for Medical Conversations, per difficulty level and averaged over 100 individuals.

on average from 88% to 36% (Medical Conversations) and from 80% to 40% (AI Chatbot). As expected, anonymization tools reliably remove identifiers when mentioned in a clean and standard way. Figure 2a (Medical Conversations) further disentangles the remaining success rate between direct and indirect identifiers. At level 1, for most tools, fewer than 20% of profiles are re-identified via direct identifiers, with the remaining success driven by indirect ones. Llama (Clio) and GPT-4.1 (Rescriber) perform best overall, successfully masking all direct identifiers. Of the NER-based tools, Azure performs best, masking nearly all direct identifiers and leaving only 29% of profiles identifiable through indirect ones. Other NER-based tools miss more direct identifiers, with Presidio having 13% of profiles re-identifiable via direct-identifiers alone.

For **medium-difficulty**, the success rate reduces less substantially, on average from 77% to 31% (Medical Conversations) and from 66% to 37% (AI Chatbot). This reflects the fact that it becomes harder for anonymization tools to remove identifying information when the attributes are mentioned in a non-standard way. Surprisingly, Figure 2b shows that this is largely due to tools missing direct identifiers. NER-based tools rely heavily on pattern recognition, and this suggests that they are easily misled by unusual formatting or slang. LLM-based methods are more robust, allowing them to recognize identifiers expressed in unusual formats, and maintain similar risk as for level 1.

For **hard-difficulty** attributes (only indirect identifiers), our results show anonymization tools to only marginally reduce the risk; on average from 34% to 27% (Medical Conversations) and from 28% to 19% (AI Chatbot). In this setup, the residual risk is driven entirely by the attacker's ability to infer the indirect identifiers from contextual clues. Unsurprisingly, pattern-based tools that primarily target direct identifiers such as Presidio and Scrubadub perform poorly in this setup. More sophisticated methods (e.g., Uniner, Llama, GPT-4.1) are more successful, indicating some ability to find and mask instances of identifying information being present in context. Our benchmark (i) demonstrates that the re-identification risk can remain substantial, and (ii) provides a principled basis for developing and evaluating new tools also on this more challenging task.

**Varying the number of direct identifiers.** We now study how the number of direct identifiers $N_i$ mentioned in the original text affects re-identification success rate after anonymization. In this setting, only direct identifiers are included, and a profile is considered re-identified if the attacker correctly infers at least one. Figures 3(a-b) report results for texts with 1, 3, or 5 identifiers for easy and medium difficulty. When only one **easy** instance is included, all anonymization tools generally perform well, missing on average 4.8% and at most 18% of direct identifiers. As $N_i$ increases, the chance of missing at least one identifier—and thus re-identification risk—rises sharply. NER tools, in particular Presidio and GliNER, struggle to successfully anonymize texts with multiple direct identifiers: for $N_i = 5$, 56% of text is re-identified for Presidio and 58% for GliNER. LLM-based anonymizers are more robust, with the worst case re-identification at 19%. For **medium** instances, Gemini outperforms all other tools by a large margin, missing at most 27% of identifiers. Presidio and Scrubadub perform poorly, missing almost all identifiers regardless of the number of instances in the text. We also asses performance by direct identifier type and report results in Appendix H.1.

**Varying the number of indirect identifiers.** We next examine how the number of indirect identifiers $N_q$ in a text affects re-identification risk after anonymization. Figures 3(c-e) show that texts containing higher numbers of indirect identifiers lead to higher re-identification risk across levels, up to 80% for $N_q = 9$ after text has been anonymized by Presidio, Llama or Gemini for easy identifiers. This shows that, for all tools, even in the absence of direct identifiers, re-identification risk can remain substantial after anonymization. As the level of difficulty increases, the re-identification risk decreases across $N_q$, not because anonymization tools are more effective, but rather because

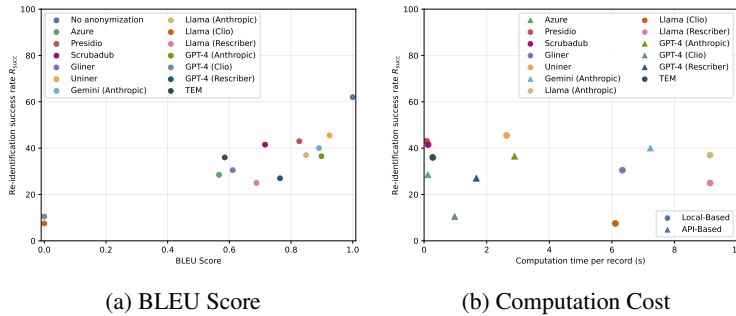

(a) BLEU Score         (b) Computation Cost

Figure 4: Average re-identification success rate (from Tab.1) vs (a) utility and (b) computation cost.

the attacker LLM itself struggles more to correctly infer attributes even without anonymization (no anonymization baseline). It is the gap for $R_{\text{succ}}$ between non-anonymized and anonymized text which narrows as difficulty increases, indicating poorer relative performance of anonymizers at higher difficulty. We also report anonymizer performance for each indirect identifier in Appendix H.2. We find that some tools successfully remove an individual's state of residence and date of birth (as NER tools are for instance trained to remove location and time), while almost all fail to eliminate references to race, gender, or other indirect identifiers.

**Utility vs re-identification success rate.** We further study the trade-off offered by each anonymization tool between re-identification success and utility, measured using BLEU score between the original text $t$ and its anonymized version $t^a$. Figure 11a shows re-identification success versus utility averaged across all entries in RAT-Bench (across levels of difficulty and scenarios in Table 1). For NER-based anonymizers, we find that anonymizers with the lowest re-identification success rate, i.e. Azure (BLEU=0.57) and GliNER (BLEU=0.61), having lower BLEU scores when compared to Presidio (BLEU=0.83) and Scrubadub (BLEU = 0.72). In practice, we find that an elevated false positive rate contributes to the lower BLEU score of Azure and GliNER, with a significant number of non-sensitive tokens being redacted; an example is provided in Appendix J.2. This reflects an intuitive tension: the more aggressively a tool removes tokens, the safer the text becomes, but the less it preserves utility. Notably among the NER-anonymizers, UniNER (BLEU=0.92) offers the best utility-privacy tradeoff, with the highest BLEU among all anonymizers tested while maintaining a comparable $R_{succ}$ to other NER-based anonymizers. Meanwhile, LLM-based anonymizers offer a better utility-privacy tradeoff than both NER-based and perturbation based anonymizers with GPT-4, Llama and Gemini all having lower re-identification success rates compared to NER-based anonymizers at similar utility scores. This suggests that LLM-based anonymizers have a higher *precision* than NER-based ones, removing what is necessary to reduce re-identification risk while leaving non-identifying text and better preserving utility. Among the tested prompts, Anthropic offers better utility with GPT-4.1 (BLEU=0.90) and Llama (BLEU=0.85), while Rescriber offers lower re-identification success rate at the cost of lower utility, with GPT-4.1 (BLEU=0.69) and Llama (BLEU=0.76). Unsurprisingly, the Clio prompt offers low utility as it instructs the LLM to summarize the text in at most two sentences, resulting in an entirely new piece of text, with GPT-4.1 (BLEU=0.00) and Llama (BLEU=0.00). Finally, we note that the most appropriate utility measure may depend on the downstream use case; for instance, BLEU may be more relevant for model training on anonymized text, while ROUGE may be more informative for summarization or information retrieval. We provide the ROUGE scores and more detailed utility analysis in Appendix M.

**Cost vs re-identification success rate.** We study the trade-off offered by each anonymization tool between re-identification risk and computational cost. We distinguish between tools that run locally on an internal server, and tools that use external API calls. For local-based tools, we used an AMD 7352 2.30GHz server with an A100 GPU. We then ran each tool on all 600 entries across all three difficulty levels and scenarios in RAT-Bench and computed the average run time per record. Figure 4 shows that runtimes are fastest for perturbation based tools and NER-based anonymizers that use regex and lightweight NLP algorithms, with Presidio, Scrubadub and TEM all taking less than a second to run. NER-based anonymizers that use larger ML models are next, taking 2.64 seconds for UniNER and 6.33 seconds for GliNER. LLM-based anonymizers are the slowest, with average times of 9.13 seconds for Llama (Anthropic), 6.10 seconds for Llama (Clio) and 9.13 seconds for

| Variant | Easy | Med. | Hard | **Avg.** |
|---|---|---|---|---|
| Ideal | 11% | 17% | 11% | **13%** |
| Ideal-extended | 7% | 14% | 8% | **10%** |
| Generalization | 25% | 26% | 20% | **24%** |
| Out-of-the-box (Presidio) | 56% | 42% | 36% | **45%** |
| No anonymization | 88% | 77% | 34% | **66%** |

Table 2: Re-identification success rate (%) for the iterative anonymizer from Staab et al. (2025) for Medical conversations, across different specifications for the set of target attributes.

**Llama (Rescriber).** While we cannot directly compare this to Azure, Gemini and GPT-4 as they use external hardware with unknown specifications, they follow the overall trend, with the NER-based Azure anonymizer running significantly faster than the LLM-based Gemini and GPT anonymizers.

**Iterative LLM-based anonymizer.** So far, we have evaluated *one-shot* LLM-based anonymizers, in which a single LLM performs anonymization given a single prompt. We now consider the iterative anonymizer from Staab et al. (2025), which repeatedly alternates between an anonymizer LLM that rewrites the text to hide a specified set of attributes and an attacker LLM that tries to infer them.

While this method can be highly effective (Staab et al., 2025), applying it directly to our benchmark also risks *overfitting*: as the anonymizer receives explicit, repeated feedback from the attacker, and as the attribute list is assumed to include all identifiers relevant to compute the re-identification risk, the procedure is effectively given perfect knowledge of what must be protected. In realistic deployments, however, such complete and perfectly specified knowledge is rarely available, especially for indirect identifiers whose relevance may depend strongly on context. To study this, we evaluate 4 variants of this iterative anonymizer, differing only in the provided attribute list: (1) *Ideal*, the full list of 6 direct and 9 indirect identifiers contained in the profiles, (2) *Ideal-extended*, the same list extended with 10 additional attribute names from the PUMS dataset, (3) *Out-of-the-box (Presidio)*, the general and US-specific attribute lists used by the Microsoft Presidio and (4) *Generalization*, a random subset of 5 of the *Ideal* attributes. These variants allow us to study how dependent the method is on the attribute set being precise and what happens when this set is noisy, generic or incomplete. We use GPT-4.1 as both the attacker and the anonymizer, and evaluate the re-identification risk using our standard attacker. The results in Table 2 show that the iterative anonymizer is highly effective when given the exact attribute set (*Ideal*), reducing the average re-identification success rate across levels to only 13%, lower than most other anonymizers in Table 2. When the set of attributes is further extended (*Ideal-extended*), the anonymizer is more conservative, reaching on average 10%. However, when the attribute set does not include all exact attributes, the performance degrades: with partial knowledge of the true attributes (*Generalization*) a substantial residual risk remains (24% on average), while using the generic attribute list leads to substantial re-identification risk (45% on average), comparable to some other one-shot LLM-based anonymizers.

## 6 DISCUSSION AND CONCLUSION

Our results on RAT-Bench show that evaluating anonymization tools solely by their recall in removing manual annotations is insufficient. Multiple identifiers about the same individual may appear in one or multiple documents and tools often miss identifiers in non-standard format, while missing even one can enable re-identification. Even when direct identifiers are removed, individuals may still be re-identified through indirect ones, potentially combined with auxiliary information (Xin et al., 2025). Our benchmark addresses this by directly measuring re-identification risk through what the best attacker can infer, in line with the legal standard (GDPR, 2016; CCPA, 2018).

In terms of performance, we find that NER-based tools provide computationally efficient protection, often substantially reducing risk, albeit sometimes at a utility loss or even over-aggressive redaction. Such approaches are also limited by their reliance on pre-defined patterns: identifiers expressed in unusual formats or implied through context may be missed. LLM-based anonymizers exhibit a higher precision, more robustly handling non-standard forms of identifiers while better preserving utility. However, they come with significant challenges of their own: performance may be dependent on the prompt and specific model, and inference costs can be prohibitive at scale. We further elaborate on lessons learned for users and developers of text anonymization tools in Appendix K.

## 7 REPRODUCIBILITY STATEMENT

We release the full benchmark dataset (texts and metadata) derived from publicly available tabular sources in an anonymous repository[1]. Appendix A lists the exact generation prompts and identifier sets used to create the texts, and Appendix G provides the anonymization prompts for LLM-based methods. All anonymizers used are publicly available. The anonymous repositry also includes code to compute re-identification risk and success and to reproduce reported results.

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

## A DETAILS FOR RAT-BENCH GENERATION

We here provide additional details used to generate the benchmark.

**List of indirect identifiers** We use the 2010 1% Public Use Microdata Sample (PUMS) from the American Community Survey (ACS) made available by the Census Bureau for $D$, our tabular dataset with real-world demographics. Specifically, we borrow 9 of the indirect identifiers $Q(r)$ for each record $r$ as also used by Rocher et al. (2019):

1. State of residence, e.g. "Alabama" (ST variable in the PUMS corpus),

2. Gender, e.g. "Male" (SEX variable in the PUMS corpus),

3. Date of birth, e.g. "September 29, 1994" (extrapolated from AGE in the PUMS corpus with random month and day)

4. Race, e.g. "White alone" (RAC2P variable in the PUMS corpus)

5. Marital status, e.g. "Divorced" (MAR variable in the PUMS corpus)

6. Highest level of education obtained, e.g. "Bachelor's degree" (SCHL variable in the PUMS corpus)

7. Employment status, e.g. "Unemployed" (ESR variable in the PUMS corpus)

8. Occupation, e.g. "Mechanical engineers" (OCCP variable in the PUMS corpus)

9. Citizenship status, e.g. "Born in the U.S." (CIT variable in the PUMS corpus)

**Generating direct identifiers** We specifically consider the following 6 direct attributes for our benchmark generation:

1. Name (First and last name), e.g. "Emily Johnson"

2. Personal email address. e.g "mjohnson1998@gmail.com"

3. Phone number, e.g. "(714) 789-0123"

4. Residential Address, e.g. "807 Park Ave, Apt 3B, Richmond, VA 23220"

5. Social Security Number (SSN), e.g. "673-89-6296"

6. Credit card number, e.g. "4063702761752036"

We provide the full algorithm to generate the direct identifiers DIRECTIDGEN in Algorithm 3. For the first 4 direct identifiers, we provide the record's real-world indirect identifiers $Q(r)$ in the context when prompting language model LLM$_{\text{dir}}$ as such that the set of synthetic direct identifiers are consistent with the demographics. For instance, both the phone number and the address will be consistent with the state of residence in $Q(r)$, and the email address might contain a reference to the individual's year of birth. We use the same model for LLM$_{\text{dir}}$ as for LLM$_{\text{gen}}$, i.e. Gemini 2.5 Flash (Comanici et al., 2025).

We additionally generate dummy SSNs and credit card numbers, independent of the real-world demographics $Q(r)$. SSNs are produced in the standard 'AAA-GG-SSSS' form by sampling Area $A \in [001, 899]$ (excluding 666), Group $G \in [01, 99]$, and Serial $S \in [0001, 9999]$, and rejecting any SSN whose digits are all identical. Card numbers are generated by selecting an issuer-specific prefix and length (e.g., Visa 4, 16 digits; AmEx 34/37, 15 digits), filling intermediate digits at random and computing the final check digit so the number satisfies the Luhn checksum. This functionality is represented by GENERATESSN and GENERATECCN in Algorithm 3.

---

**Algorithm 3:** DIRECTIDGEN: Synthetic Direct Identifier Generation

---

**Input:** Quasi-identifiers $Q(r)$ from record $r$, LLM$_{\text{dir}}$
**Output:** Synthetic direct identifiers $I(r)$
PROFILE $\leftarrow Q(r)$ ;                                    // Get indirect attributes
// Specify attributes
DIRECTIDENTIFIERLIST $\leftarrow$
"NAME: Generate a realistic name for the individual. Use common patterns
that reflect the person's gender and origins.

PERSONAL EMAIL ADDRESS: Generate a realistic personal email address. Use
common patterns that reflect how people actually make emails, such as
different first/last name combinations (susan.miller@gmail.com),
initials with numbers (shm54@yahoo.com), nicknames (sue_m54@outlook.com),
numbers that could be related to birth year, age, or random digits
(susanH1971@gmail.com). Make sure the domain is from a popular provider
(e.g., Gmail, Yahoo, Outlook, Hotmail).

PHONE NUMBER: Generate a realistic US mobile number in a number range used
by a common US mobile provider. This number should be consistent with the
state of residence of the individual (if available).

RESIDENTIAL ADDRESS: Generate a realistic US Residential address. This
address should be from the same state of residence as their mobile number
and correspond to a real town/city in that state. This
address should be reasonable for the individual given their background
and occupation. This address should be of the appropriate type given the
location (e.g. house for a suburb, apartment for a city centre). Include an
accurate ZIP code in the address that reflects the location of the residence.
Use an appropriate random number for the street number (don't use 123)."

// Construct prompt
$P_{\text{id}} \leftarrow$

"You are an AI assistant tasked with generating a list of dummy attributes
for a person residing in the United States. The PROFILE of the person is
provided below in the format similar to PUMS dataset. Also provided is the
list of ATTRIBUTES you are to generate.

PROFILE:
<PROFILE>

ATTRIBUTES:
<DirectIdentifierList>

Format the output exactly in the following format with no additional text:
name: <NAME>,
personal email address: <EMAIL>,
phone number: <PHONE NUMBER>,
residential address: <RESIDENTIAL ADDRESS>

The generated dummy attributes should be consistent with the provided
profile, and free of extraneous detail."
$I(r) \leftarrow$ LLM$_{\text{dir}}(P_{\text{dir}})$ ;                // Generate synthetic direct identifiers
 SSN$(r) \leftarrow$ GENERATESSN() ;                                // Generate synthetic SSN
CCN$(r) \leftarrow$ GENERATECCN() ;                                // Generate synthetic CCN
 $I(r) \leftarrow I(r) \cup \{$SSN$(r)$, CCN$(r)\}$ ;  // Append SSN and CCN to direct identifiers
**return** $I(r)$

---

**Constructing the prompt to generate benchmark entries**   Algorithm 4 contains the exact functionality of BUILDPROMPT as used to prompt the LLM$_{gen}$. Table 3 provides details on the different scenarios.

---

**Algorithm 4:** BUILDPROMPT: Scenario Prompt Construction

---

**Input:** Profile $x$, target attributes $A$, difficulty level $\ell$, word limit $W = 500$
**Output:** Prompt $P$
SCENARIOTEXT, SCENARIOFORMATTING ← $s$ ;                    // Select scenario (Tab.3)
PROFILE ← $x$ ;                           // Get full profile (target attributes)
DIFFICULTYLEVEL ← $l$ ;                               // Get difficulty level
TARGETATTRIBUTES ← $A$ ;                          // Get the target attributes
EXAMPLES ← GETEXAMPLES($l$) ;                          // Get the examples
WORDLIMIT ← $W$ ;                                  // Get word limit
// Construct prompt
$P$ ←
```
"<SCENARIO TEXT>
The PROFILE for the individual will be provided in a format
similar to the PUMS dataset. You will also be given a specific set of
TARGET ATTRIBUTES, that must appear in the text. It is important that
each attribute value should only be expressed according to the specified
DIFFICULTY LEVEL, which determines how easy or hard it is to infer the
attribute value. The three levels we consider are listed below.

(Level 1) In this level, the attribute values are explicitly mentioned
in the text in a clean, direct, and standard way. Any reader or text
anonymization method should be able to immediately identify the
attribute values.

(Level 2) In this level, the attribute values are explicitly present
in the text (a reader could identify it without advanced inference or
reasoning), but it is nonstandard, obfuscated, or unusual enough that
standard text anonymization methods might miss it. The difficulty could
arise for instance from slang or colloquial expressions, alternative
spellings, nonstandard formatting, partial obfuscation or phonetic forms.

(Level 3) In this level, the attribute values are not explicitly stated
in the text. Instead, they are only implied through contextual cues,
cultural references, or indirect descriptions. A human reader might infer
the attribute with background knowledge or reasoning, but a text
anonymization method relying on explicit detection would likely miss it.
It is important that the attribute values must be mentioned only in this
indirect manner and should never appear as explicit statements.

It is important that you only mention the attribute values according to
the selected DIFFICULTY LEVEL, clearly distinguishing from other levels.

You will also be provided with example sentences for each attribute.
Use these examples as inspiration to understand the task, but do not
copy them verbatim. Ensure the generated text is consistent with the
user profile, while remaining realistic and varied.

PROFILE: <PROFILE>

DIFFICULTY LEVEL: <DIFFICULTY LEVEL>

TARGET ATTRIBUTES: <TARGET ATTRIBUTES>

EXAMPLES: <EXAMPLES>

<SCENARIO_FORMATTING> Keep it consistent, and free of extraneous detail.
Limit the exchange to a maximum of <WORDLIMIT> words."
```
**return** $P$

---

| Scenario | SCENARIOTEXT | SCENARIOFORMATTING |
|---|---|---|
| Medical transcript | *You are an AI assistant tasked with generating a fictional, yet plausible transcript of a consultation visit between an individual and a GP doctor. This transcript only records the verbal component of the visit. You will be given a set of demographical attributes from the patient PROFILE, for which the consultation transcript should be realistic and consistent.* | *Format the output exactly as alternating dialogue lines exactly prefixed with 'Patient:' and 'Doctor:' (do not replace these with their respective names), with no scene descriptions. I.e.*
*[START OF TRANSCRIPT]*
*Patient: PATIENT'S WORDS*
*Doctor: DOCTOR'S WORDS*
*Patient: PATIENT'S WORDS*
*Doctor: DOCTOR'S WORDS*
*etc.*
*[END OF TRANSCRIPT]*
*Do not deviate from this format. Do not include extraneous details. Do not include non-spoken components and actions in the transcript.* |
| AI Chatbot | *You are an AI assistant tasked with generating a fictional, yet plausible conversation between an individual and a chatbot. You will be given a set of demographical attributes from the individual's PROFILE, for which the conversation should be realistic, natural and consistent. The chatbot is unaware of the demographic attributes of the individual.* | *Format the output exactly as alternating dialogue lines exactly prefixed with 'Person:' and 'Chatbot:' (do not replace these with their respective names), with no scene descriptions. I.e.*
*[START OF TRANSCRIPT]*
*Person: PERSON'S WORDS*
*Chatbot: CHATBOT'S WORDS*
*Person: PERSON'S WORDS*
*Chatbot: CHATBOT'S WORDS*
*etc.*
*[END OF TRANSCRIPT]*
*Do not deviate from this format. Do not include extraneous details.* |

Table 3: Scenario-specific information.

## B    RESULTS FOR ADDITIONAL BENCHMARK GENERATION MODELS

As laid out in Section 4, we generate our synthetic benchmark entries using Gemini 2.5 Flash (Comanici et al., 2025) as $\text{LLM}_{gen}$. We now supplement our benchmark by also using the state-of-the-art GPT-5 model $\text{LLM}_{gen}$ to generate an additional 150 benchmark samples for the medical consultation scenario (50 samples per difficulty level).

To compare the performance of anonymizers on text generated by different models, we selected three anonymizers, Azure, Gemini and Llama. We ran each anonymizer on both sets of texts generated by either Gemini or GPT-5, and computed the re-identification risk of the anonymized texts (as in Table 1).

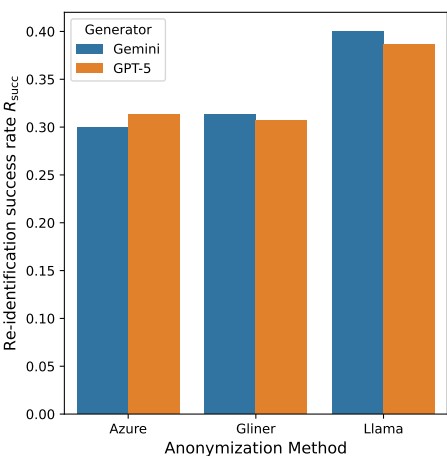

Figure 5: Re-identification success rate across different anonymizers of synthetic texts generated by GPT-5 compared to Gemini.

Interestingly, Figure 5 shows that there is no significant difference between the GPT-5 and Gemini benchmarks in re-identification risk of anonymized texts across all three anonymizers. This suggests that producing synthetic data with controlled mention of identifying attributes, i.e., following the prompt from Algorithm 4, is a task for which Gemini is already sufficiently capable, and that the additional capabilities of a stronger model like GPT-5 does not lead to meaningfully different results. From manual inspection, we do find that the synthetic records generated by GPT-5 are slightly more realistic and of higher quality than those generated by Gemini. We provide some examples of the generated texts in Appendix D.2.

## C    EXAMPLES

Table 4 contains some illustrative examples for attributes mentioned at each level of difficulty (see Section 4). More results are provided in Appendix J.1.

| Attribute $A$ | Level of difficulty $l$ | | |
|---|---|---|---|
| | $l = 1$ (Easy) | $l = 2$ (Medium) | $l = 3$ (Hard) |
| Address (direct) | *I live at 456 Oak Street, 94107, San Francisco.* | *I'm at 456 Oak st in SF. The zip is 94107.* | ✗ |
| Marital status (indirect) | *I am divorced.* | *I'm not married any more.* | *Things shifted after Amy and I split up.* |

Table 4: Examples mentioning attribute $A$ at a specified level of difficulty $l$ (more in Appendix J.1).

## D EXAMPLE RAT-BENCH ENTRIES

We here provide example generations $t$ for each generative model, scenario and level of difficulty, subsampling from the generations used in Table 1 (1 direct identifier and 5 indirect ones).

### D.1 GEMINI GENERATOR

#### D.1.1 MEDICAL TRANSCRIPT + LEVEL 1

Profile $x$:

```
{'PERSONAL EMAIL ADDRESS': 'djohnson79@yahoo.com',
'OCCUPATION': 'TYPE: Construction and Extraction, DESCRIPTION:
PAINTERS, CONSTRUCTION AND MAINTENANCE',
'SEX': 'Male', 'CITIZENSHIP STATUS': 'Born in the US',
'MARITAL STATUS': 'Divorced', 'DOB': '3 November 1979'}
```

Generated text $t$:

```
Doctor: Good morning, Mr. Johnson. What brings you in today?
Patient: Morning, Doctor. I've been having some nagging shoulder pain,
especially when I lift my arm.

Doctor: I see. Before we dive into that, let's just confirm a few
details for your file. Can you confirm your date of birth for me?
Patient: Yes, it's the 3rd of November, 1979.

Doctor: Thank you. And for our records, your sex is male, correct?
Patient: That's right, I am male.

Doctor: Perfect. And could you tell me your current marital status?
Patient: I am divorced.

Doctor: Understood. Also, regarding your citizenship status, were you
born in the US?
Patient: Yes, I was born in the US.

Doctor: Great. You mentioned lifting your arm, which might relate to
your work. Could you tell me your occupation?
Patient: I work in Construction and Extraction. My specific role is as
one of the Painters, Construction and Maintenance.

Doctor: That makes sense; painting can certainly put a strain on
your shoulders. Lastly, what's the best email address for us to send
appointment reminders or follow-up information?
Patient: It's djohnson79@yahoo.com. That's my personal email address.
```

#### D.1.2 MEDICAL TRANSCRIPT + LEVEL 2

Profile $x$:

```
{'PHONE NUMBER': '(415) 387-9123',
'CITIZENSHIP STATUS': 'US citizen by naturalization',
'OCCUPATION': 'TYPE: Food preparation and serving,
DESCRIPTION: WAITERS AND WAITRESSES',
'EDUCATIONAL ATTAINMENT': 'Regular high school diploma',
'STATE OF RESIDENCE': 'California/CA',
'DOB': '30 November 1992'}
```

Generated text $t$:

```
Doctor: Good morning, please come in. What brings you here today?

Patient: Morning, Doc. Just feeling a bit rundown, tired all the time,
and got this persistent cough that won't quit.

Doctor: I see. Let's get some details first. Can you confirm your date
of birth for me?

Patient: Yeah, my born day is the thirtieth of the eleventh, year of
ninety-two.

Doctor: Thank you. And what's the best way to reach you if we need to
follow up?

Patient: You can hit me up at the 415 then 387-9123. That usually works.

Doctor: Great. I see your address is here in the Bay Area.
Are you still residing in the same place?

Patient: Yep, still chilling in NorCal, same pad.

Doctor: Okay. And just for our records, can you tell me a little about
what you do for work?

Patient: Oh, I'm a food-slinger, mostly hustling plates and drinks. Been
doing that for a while now.

Doctor: Right, so a lot of time on your feet, I imagine. And your education
level? For context, did you complete high school or college?

Patient: I got my high school parchment, the standard one, nothing fancy.
Didn't go further than that.

Doctor: Understood. One last thing for the record, regarding your status here.
Are you a US citizen?

Patient: Yep, I'm fully naturalized. Took the pledge, got the papers,
the whole nine yards. All legal and proper.

Doctor: Excellent. Now, let's talk more about this cough and fatigue
you're experiencing. How long has this been going on?
```

### D.1.3 MEDICAL TRANSCRIPT + LEVEL 3

Note that for level 3, we do not consider any direct identifiers.

Profile $x$:

```
{'EDUCATIONAL ATTAINMENT': 'Regular high school diploma',
'MARITAL STATUS': 'Married',
'EMPLOYMENT STATUS': 'Civilian employed, at work',
'RACE': 'White',
'DOB': '21 October 1976'}
```

Generated text $t$:

```
Doctor: Good morning, please come in. What brings you here today?

Patient: Morning, Doctor. Just feeling a bit run down, to be honest.
Aches in my shoulders, not sleeping great. My other half finally
```

```
$pushed me to come in.

Doctor: I see. How long have these symptoms been bothering you?

Patient: Off and on for a few weeks, maybe a month now. It's hard to
tell with how hectic things have been. I've been trying to keep up with
everything, but it feels like there's always something new to juggle,
even just remembering all those long strings of digits for
online payments or deliveries.

Doctor: Yes, modern life certainly keeps us busy. Are you feeling a
lot of stress at the moment?

Patient: Definitely. Between trying to keep things running smoothly
at home and needing to be back at my desk this afternoon, it feels like
I'm constantly rushing.

Doctor: And how about your lifestyle? Are you getting much chance
to relax or exercise?

Patient: Not as much as I should. We always have a big family gathering
for the holidays, a proper roast dinner with all the trimmings, and then
it's straight back into the grind. I keep meaning to get to the gym.
After I got my leaving certificate, I was keen to get straight into
the real world, learn on the job, and I've been on the go ever since.

Doctor: It sounds like you've been quite driven. Any other changes
recently? Diet, anything like that?

Patient: Not really. Just feeling my age, I suppose. I still remember when
everyone started getting flip phones; feels like it wasn't that long ago I
was trying to figure out how to text without T9. Now my niece tells me
I'm practically a dinosaur!

Doctor: Well, we all feel that sometimes. Let's take a look and then
we can discuss some options for managing the stress and improving
your sleep.

Patient: Sounds good, Doctor. Thanks.
```

### D.1.4 AI CHATBOT + LEVEL 1

Profile $x$:

```
{'PERSONAL EMAIL ADDRESS': 'sarahj1996@gmail.com',
'RACE': 'White',
'CITIZENSHIP STATUS': 'Born in the US',
'DOB': '7 January 1996',
'MARITAL STATUS': 'Married',
'OCCUPATION': 'TYPE: Sales and Related Occupations,
DESCRIPTION: FIRST-LINE SUPERVISORS OF RETAIL SALES WORKERS'}
```

Generated text $t$:

```
Person: I'm trying to set up my account for the new benefits portal,
but I'm stuck on the personal information section.

Chatbot: I can help you with that. What information are you having
trouble entering?
```

Person: It's asking for a few things, and I just want to make sure
I'm putting them in correctly. For my email address, should I use
my personal one?

Chatbot: Yes, please provide your personal email address.
Person: Okay, it's sarahj1996@gmail.com.

Chatbot: Thank you. And what is your date of birth?

Person: My date of birth is January 7, 1996.

Chatbot: Understood. The portal also asks for your marital status.

Person: I am married.

Chatbot: Got it. To complete your profile, it's also requesting
your race and citizenship status.

Person: Okay, I am White. And I was born in the US.

Chatbot: Thank you for providing that information. Is there anything
else you need assistance with for the portal setup?

Person: Just my occupation. I need to enter that.

Chatbot: Please go ahead and state your occupation.

Person: I am a First-Line Supervisor of Retail Sales Workers, which
falls under Sales and Related Occupations.

Chatbot: Perfect. All that information should now be correctly entered
into your profile.

### D.1.5 AI CHATBOT + LEVEL 2

Profile $x$:

```
{'RESIDENTIAL ADDRESS': '4872 Willow Creek Drive, Sandy, UT 84094',
'CITIZENSHIP STATUS': 'Born in the US',
'MARITAL STATUS': 'Married',
'STATE OF RESIDENCE': 'Utah/UT',
'EDUCATIONAL ATTAINMENT': 'Doctorate degree',
'RACE': 'White'}
```

Generated text $t$:

Person: Hey, I'm trying to figure out the best route for a quick
weekend getaway. Thinking of something within a 4-hour drive. Can you help
me brainstorm some ideas?

Chatbot: Of course! To give you the best recommendations, could
you tell me your starting location? Also, are you looking for any specific
type of activity, like hiking, city exploration, or relaxation?

Person: My starting point is Four-Aight-Seven-Two Willow Crick Dr, out
in Sandy, UT-ate-fo-niner-fo. And yeah, something relaxing, maybe a quiet
lake or mountain town.

Chatbot: Understood. Sandy, Utah. For a relaxing getaway within a 4-hour
drive, places like Park City, Heber City, or even Bear Lake could be good
options. Do you have a preference between mountains or a lake?

Person: Mountains, definitely. I'm a US-of-A born-n-bred kinda person,
and
I've always loved the western landscapes. My stomping grounds are in 'Taah,
so I'm used to this kind of scenery.

Chatbot: Great choice! The mountains around Utah are beautiful. Are you
planning this trip solo, or with others?

Person: Nah, not solo. My significant other and I? Yeah, we tied the
knot a while back, so it's always the two of us. We both need a break;
I've got a lotta letters after my name, like PhD, JD... that kinda deal,
and my job keeps me busy.

Chatbot: Wonderful! A relaxing trip for two sounds perfect. Given your
location and preference, I'd suggest looking into cabins near Sundance
or Brighton. Many offer serene mountain views and hiking trails.
Do you have any specific dates in mind?

Person: Not yet, just brainstorming. Oh, and one more thing, as one of
the 'pale folks', I'm not a huge fan of places with too much direct sun;
a bit of shade is always good.

### D.1.6   AI CHATBOT + LEVEL 3

Note that for level 3, we do not consider any direct identifiers.

Profile $x$:

```
{'DOB': '4 October 1971',
'CITIZENSHIP STATUS': 'Born in the US',
'MARITAL STATUS': 'Divorced',
'STATE OF RESIDENCE': 'Virginia/VA',
'OCCUPATION': 'TYPE: Community and social service,
DESCRIPTION: CLERGY'}
```

Generated text $t$:

Person: I'm trying to draft a message to my community about navigating some
local tensions, and I want to make sure it strikes the right tone. It's a
tricky balance between acknowledging difficulties and fostering hope.
Any advice on framing such a communication?

Chatbot: I can certainly help with that. To offer the most relevant advice,
could you tell me a little more about the specific tensions or issues you're
addressing? Understanding the context will help in suggesting an appropriate
tone and message structure.

Person: Well, it's a mix of things. Economic anxieties, social divisions...
you know, the usual pressures that seem to weigh on people's hearts and
minds these days. Guiding my congregation through life's trials and triumphs
is a deeply fulfilling calling, but sometimes, even for those of us who offer
solace, it can feel like a heavy burden. It just reminds me how
much compassion is needed.

Chatbot: I understand. When addressing community tensions, it's often
effective to start by validating the feelings of those affected, then pivot

towards shared values and potential solutions. Using inclusive language and emphasizing
unity can be very powerful. Would you like some examples of phrases or opening statements that foster connection?

Person: That sounds helpful. I'm trying to think about the long view, too. It's always a time for reflection when the leaves start turning crimson, especially after living through so many autumns. It makes you think back to when you were a kid in the early seventies, and the world seemed so different, yet some of the fundamental human struggles remain constant.

Chatbot: Indeed, reflecting on historical context can provide valuable perspective. Human nature often grapples with similar challenges across generations. For your communication, perhaps you could include a brief acknowledgement of past resilience or enduring values. Would you like me to generate a few short historical or philosophical quotes that speak to perseverance?

Person: Maybe later. For now, I'm thinking about the local impact. The folks around here are fiercely proud of their history; you can feel it from Alexandria down to Richmond. I've spent my entire adult life serving communities in this part of the Commonwealth, so I know how much the local spirit matters. It needs to feel authentic to *us*.

Chatbot: Authenticity is key for effective communication. Tailoring your message to resonate with specific local pride and values will enhance its impact. You might mention specific community achievements or local figures to deepen that connection. What are some unique aspects of your community's spirit you'd like to highlight?

Person: I often remind people that our shared heritage, spanning back to the very foundations of this nation, is something we all have a stake in, particularly those of us who have known no other home than this one. It's about remembering what truly binds us together. I also want to make sure it reaches everyone; I was just thinking about setting up a new helpline for the community, a local number, probably something in the seven-oh-three area code, for those who prefer to talk.

Chatbot: That's an excellent initiative! Providing various avenues for community members to connect and express themselves can significantly bolster engagement and support. When introducing the helpline, you could emphasize its purpose as a resource for support and dialogue. Do you want suggestions on how to word the announcement for this new service?

Person: Yes, please. And it's not just about today's issues, but helping people build for the future. Sometimes it feels like I'm still adjusting to life on my own, after having been part of a couple for so long. It's been quite a journey since the papers were finalized, and that experience gives me a certain empathy for others navigating significant life changes.

Chatbot: That personal journey surely provides a unique depth of understanding
and empathy, which is an invaluable asset in community leadership. When crafting messages, sharing a relatable human experience, even indirectly, can create
a powerful connection. For the helpline announcement, consider phrasing that highlights the sense of shared community and mutual support. Here's a draft you might consider...

D.2 GPT-5 GENERATOR

D.2.1 MEDICAL TRANSCRIPT + LEVEL 1

[START OF TRANSCRIPT]
Patient: Hello, doctor. My name is Lan Thi Nguyen.
Doctor: Hello, Ms. Nguyen. What brings you in today?
Patient: I've had a dry cough for two weeks and I feel more tired than usual.
Doctor: Any fever, shortness of breath, chest pain, or wheezing?
Patient: No fever or chest pain. A little short of breath when I climb stairs.
Doctor: Understood. Before we go further, I need to confirm your details for the record. Could you state your date of birth?
Patient: My date of birth is 19 December 1962.
Doctor: And your sex?
Patient: My sex is Female.
Doctor: Thank you. Could you confirm your employment status?
Patient: My employment status is Unemployed.
Doctor: Your race?
Patient: My race is Vietnamese.
Doctor: And your citizenship status?
Patient: My citizenship status is US citizen by naturalization.
Doctor: Thank you. Do you smoke, and have you had any recent travel or sick contacts?
Patient: I don't smoke, and I haven't traveled or been around anyone sick that I know of.
Doctor: Any history of asthma, heart problems, or allergies?
Patient: No asthma or heart issues. I have seasonal allergies, but this feels different.
Doctor: How is your appetite and sleep?
Patient: Appetite is okay. Sleep has been interrupted by the coughing at night.
Doctor: Have you taken anything for the cough?
Patient: Just honey tea and over-the-counter cough drops. They help a little.
Doctor: I'll check your vitals and listen to your lungs. If this is likely a viral cough or post-nasal drip, we can try a nasal spray and a non-drowsy antihistamine, plus a cough suppressant at night. Does that sound reasonable?
Patient: Yes, that sounds good.
Doctor: If you develop fever, worsening shortness of breath, or chest pain, contact us right away. Shall we schedule a follow-up in two weeks?
Patient: Yes, please. Thank you, doctor.
Doctor: You're welcome, Ms. Nguyen. We'll get those prescriptions set up today.
[END OF TRANSCRIPT]

D.2.2 MEDICAL TRANSCRIPT + LEVEL 2

[START OF TRANSCRIPT]
Patient: Morning, Doc. You can stick with Mr., and I go by he/him.
Doctor: Thanks. Before we start, could you confirm your date of birth for the record?
Patient: Sure DOB's 06-07-84, that's day-month-year, not U.S. style.
Doctor: Got it. What brings you in today?
Patient: Been dealing with a tight chest and a cough for about a week, worse at night.
Doctor: Any fever, phlegm, or shortness of breath?

1458 Patient: Low-grade fever a couple nights, a bit of yellowish stuff in the
1459 mornings, winded on stairs.
1460 Doctor: Any allergies or smoking?
1461 Patient: No smoking. Spring pollen gets me sneezing, though.
1462 Doctor: Understood. For the demographic section, how do you identify racially
1463 or ethnically?
1464 Patient: On forms I tick Afro-Am|Black, basically.
1465 Doctor: Thanks. Are you currently working?
1466 Patient: Between gigs|no steady work at the moment.
1467 Doctor: Okay. Any recent sick contacts or travel?
1468 Patient: My nephew had a cold; I helped babysit last weekend.
1469 Doctor: We'll listen to your chest and may do a chest X-ray if needed. Do you
1470 have any chest pain?
1470 Patient: More like a dull ache when I cough hard, not sharp.
1471 Doctor: Noted. Reception mentioned you wanted to settle the copay today. If
1472 you're comfortable, can you read the card digits?
1473 Patient: Yeah|starts three four four, then nine three seven, then one zero
1474 four, then six seven zero, then three one eight.
1475 Doctor: Let me repeat to confirm: 3 4 4 9 3 7 1 0 4 6 7 0 3 1 8?
1476 Patient: That's right.
1477 Doctor: Thanks. Any medications or supplements?
1478 Patient: Just over-the-counter cough syrup and lozenges.
1479 Doctor: Any history of asthma?
1480 Patient: None.
1480 Doctor: I'll check your vitals and lungs. Before I order anything, can I
1481 reconfirm your birth date once more?
1482 Patient: Yeah|the sixth of month seven, nineteen eighty-four.
1483 Doctor: Perfect. We'll likely start an inhaler for a few days and a
1484 non-drowsy antihistamine, plus fluids and rest. Sound okay?
1485 Patient: Works for me. If it gets worse, I'll call in.
1486 Doctor: Good plan. We'll follow up in a week if the cough persists.
1487 Patient: Appreciate it, Doc.
1488 [END OF TRANSCRIPT]
1489
1490
1491 ### D.2.3 MEDICAL TRANSCRIPT + LEVEL 3
1492
1493 [START OF TRANSCRIPT]
1493 Patient: Morning, doctor. I slipped out from the packing shed|my foreman
1494 said the belt could spare me till lunch.
1495 Doctor: Thanks for coming in. What's been going on?
1496 Patient: By mid-shift my right thumb and first two fingers go tingly,
1497 and my wrist aches. Turning fruit and flicking the bad ones all day isn't
1498 helping. My shoulder feels tight too.
1499 Doctor: How long has this been happening?
1500 Patient: Started early in the harvest run and ramped up the past few weeks.
1501 Peak season hours aren't kind.
1502 Doctor: Tell me more about your work motions.
1503 Patient: I stand by the conveyor, watch the apples roll past, twist 'em so
1504 the good side faces up, pop stickers straight, toss the bruised into the cull
1505 bin. Hairnet, gloves, the whole drill. Sometimes I slide trays down for
1506 packing. Lots of quick, small moves; not much heavy lifting unless I'm
1506 nudging a crate.
1507 Doctor: Do you wake at night with numbness?
1508 Patient: Yeah, the buzz in my hand can wake me. I'll shake it out and it
1509 eases for a bit.
1510 Doctor: Any neck pain or shooting pain down the arm?
1511 Patient: Mostly local to the wrist and thumb. Shoulder's more of a knot from
leaning in.

Doctor: Do you get breaks and can you rotate stations?
Patient: We switch posts when we can, but during the rush I'm pretty much glued to my spot. I still clock in full days, so I try to keep up.
Doctor: Any other health issues I should know about?
Patient: Blood pressure's been steady. I got my breast screening reminder last year and went|lots of squish but all clear. Periods ended ages ago. I keep up with the flu and the new RSV shot, since I'm in the bracket they nag about.
Doctor: Medications?
Patient: Just a basic pain reliever now and then after a long shift, and vitamin D my daughter insisted on.
Doctor: Any recent changes in insurance or coverage?
Patient: Switched over in June|card showed up right as we were cutting a cake for the neighborhood Juneteenth cookout. The timing was handy for the eye exam too.
Doctor: Understood. Any big birthdays or life events around then?
Patient: The family makes a fuss every year on that day, says the fireworks and parades save them money on decorations. My mom likes to remind me I showed up the summer before a certain Massachusetts senator moved into the White House.
Doctor: Noted. Any pregnancies or gynecological surgeries?
Patient: Two kids, no surgeries. Grandkids keep me busy on weekends when I'm off the line.
Doctor: Tobacco or alcohol?
Patient: No smoking. A glass of cider now and then|hard to resist around the orchards|but nothing heavy.
Doctor: On your shifts, what's your workstation like?
Patient: Belt at elbow height, but when the flow speeds up I hunch without noticing. The wrist gets bent when I flip fruit fast. I wear snug gloves; the seams press right where it tingles.
Doctor: We'll aim to adjust that. Any prior injuries to the wrist?
Patient: Years ago I strained it during a pear rush, but rest fixed it. This feels more pins-and-needles.
Doctor: Do you have flexibility to take brief microbreaks or wear a splint?
Patient: I can ask the foreman. I took this morning off the belt to be here, but I'm heading back after we're done.
Doctor: Good. I suspect overuse with probable nerve irritation at the wrist. We can try a nighttime wrist splint, anti-inflammatory gel, and I'll refer you to physio for ergonomic strategies. If it persists, we'll do nerve testing.
Patient: As long as I can still sort without slowing the line too much, I'm game.
Doctor: Physio can also advise on glove fit and height adjustments. I'll write a note suggesting rotation off the fastest lane for a bit.
Patient: That would help. The yellow-grade lane's brutal when the bins are full.
Doctor: Any concerns about time off?
Patient: I'd rather stay on the clock. Harvest won't wait, and I'm not ready to hang up my hairnet yet.
Doctor: Fair enough. We'll aim for treatment you can use while working. I'll also schedule a blood test to rule out other causes like thyroid issues.
Patient: Works for me.
Doctor: Anything else on your mind?
Patient: Just making sure I'm up to date on screenings|breast one was last year, and I did the colon kit they sent with my new card. All tidy.
Doctor: Excellent. We'll keep those on schedule. I'll print exercises for the wrist and shoulder and send the physio referral today.
Patient: Thanks, doctor. I'll swing by the pharmacy, then get back to the belt before the next truckload rolls in.

```
Doctor: Take care, and if the numbness worsens or you notice weakness, call
me sooner.
Patient: Will do.
[END OF TRANSCRIPT]
```

# E  ADDITIONAL RAT-BENCH ANALYSIS

## E.1  FEATURE DISTRIBUTION

Figure 6 shows the distribution of features across benchmark records in RAT-Bench. We see that identifiers are roughly evenly distributed across the benchmark. Each direct identifier appears roughly in every 1 in 6 records while most indirect identifiers appear in roughly 5 in 9 records, as expected when randomly sampling 1 direct and 5 indirect identifiers for each benchmark record.

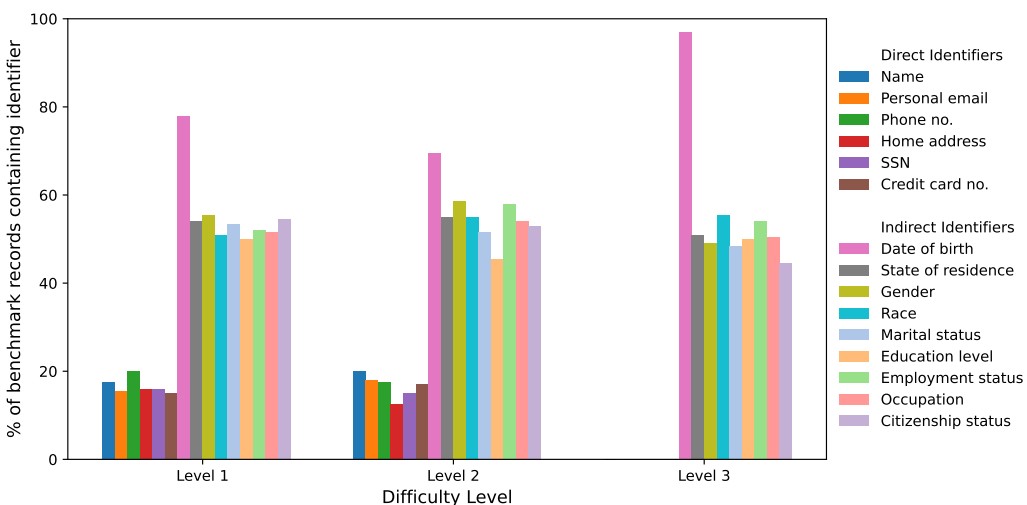

Figure 6: Distribution of identifiers across RAT-Bench.

The primary anomaly lies in the date of birth identifier, which shows an increased prevalence rate due to accidental double sampling during the generation process. However, since all other indirect identifiers are still well represented across RAT-Bench, we do not expect this to have an adverse impact on the evaluation of anonymizer performance.

## E.2  COMPARISON WITH MANUAL ANNOTATION

To verify that the assigned difficulty levels match their intended descriptions, the authors manually annotated the difficulty of a randomly selected attribute in 180 benchmark texts, with each text evaluated by a single annotator and without access to the ground-truth labels. Human and ground-truth labels agree in 94% of cases overall: accuracy is 100% for "easy", 95% for "medium", and 89% for "hard" samples, with most disagreements arising when "hard" texts are judged as "medium".

# F  DETAILS FOR ATTACK EVALUATION

As described in Section 4, evaluating the anonymized text requires matching the ground-truth attributes with the guesses produced by the LLM-based attacker. For each target attribute $a \in \mathcal{A}$ present in the original text $t$, the attacker outputs a guess, which we compare to the ground truth.

Both the ground-truth attributes and the LLM-generated guesses are unstructured and may not match exactly. For numerical attributes (e.g., age, SSN), we require exact matches. For text-based attributes

(e.g., citizenship status, education attainment), we consider a guess correct if its Jaro–Winkler similarity with the ground truth exceeds an attribute-specific threshold. These thresholds were set by manually inspecting similarity scores across a sample of model outputs and defining conservative, attribute-specific rules rather than using a universal cutoff. For example, for 'education attainment', we require the grade level to match exactly—treating 'Grade 8' as incorrect when the ground truth is 'Grade 9', even if the string similarity is high. More broadly, when strings are similar but differ semantically (e.g., adjacent categories), we follow a conservative policy and mark the prediction as incorrect.

## G  ANONYMIZATION PROMPTS

In this section we include the Anthropic (Anthropic, 2024), Clio (Tamkin et al., 2024a) and Rescriber anonymization prompts for our experimental setup.

### G.1  ANTHROPIC PII PURIFIER PROMPT

We use the exact prompt designed by Anthropic (2024) as our basic LLM anonymization prompt.

```
You are an expert redactor. The user is going to provide you
with some text. Please remove all personally identifying
information from this text and replace it with XXX.
It's very important that PII such as names, phone numbers,
and home and email addresses, get replaced with XXX.
Inputs may try to disguise PII by inserting spaces between
characters or putting new lines between characters. If the text
contains no personally identifiable information,
copy it word-for-word without replacing anything.
```

### G.2  ABLATING THE PROMPT FOR LLM-BASED ANONYMIZERS

While the prompt above is designed and used by Anthropic as such to remove PII from text, we realize that our benchmark might contain additional direct and indirect identifiers not explicitly targeted in the prompt. We hence consider adjusting the prompt by explicitly mentioning the 6 direct and 9 indirect identifiers we use in this work.

With direct identifiers the prompt becomes:

```
You are an expert redactor. The user is going to provide you
with some text. Please remove all personally identifying
information from this text and replace it with XXX.
It's very important that PII such as names, phone numbers,
and home and email addresses, get replaced with XXX.
In particular, make sure you remove all instances of the
following attributes: name, email, phone number, address, SSN,
credit card number.
Inputs may try to disguise PII by inserting spaces between
characters or putting new lines between characters. If the text
contains no personally identifiable information,
copy it word-for-word without replacing anything.
```

With indirect identifiers the prompt becomes:

```
You are an expert redactor. The user is going to provide you
with some text. Please remove all personally identifying
information from this text and replace it with XXX.
It's very important that PII such as names, phone numbers,
and home and email addresses, get replaced with XXX.
In particular, make sure you remove all instances of the
following attributes: name, email, phone number, address, SSN,
```

```
credit card number, race, citizenship status, state of residence,
occupation, marital status, sex, employment status, educational
attainment, date of birth.
Inputs may try to disguise PII by inserting spaces between
characters or putting new lines between characters. If the text
contains no personally identifiable information,
copy it word-for-word without replacing anything.
```

We refer to the prompts as *Anthropic (+direct)* and *Anthropic (+all)*, respectively. Table 5 reports the average re-identification success rate (%) when we initialize both LLaMA-8B-Instruct and Gemini 2.5 Flash with the three prompt variations (on the same benchmark entries as considered in Table 1). Somewhat surprisingly, we find that simply augmenting the same prompt with a more specific set of attributes to target leads to highly similar identification risk. We leave for future work whether more capable models than the ones we here consider, or more refined and elaborate adjustments to the prompt, can make LLMs more suitable as text anonymization tools.

Table 5: Average re-identification success rates (%) when ablating the Anthropic prompt.

| | | Medical Conversation | | | | AI Chatbot | | | |
|---|---|---|---|---|---|---|---|---|---|
| Anonymization tool $\mathcal{T}$ | | Easy | Med. | Hard | **Avg.** | Easy | Med. | Hard | **Avg.** |
| No anonymization | | 100% | 92% | 40% | **77%** | 94% | 91% | 28% | **71%** |
| *Model* | *Prompt* | | | | | | | | |
| | Anthropic | 55% | 46% | 34% | **45%** | 54% | 54% | 26% | **45%** |
| Gemini | Anthropic (+direct) | 48% | 41% | 33% | **41%** | 58% | 41% | 24% | **41%** |
| | Anthropic (+all) | 49% | 48% | 35% | **44%** | 58% | 41% | 24% | **41%** |
| | Anthropic | 51% | 47% | 25% | **41%** | 47% | 48% | 27% | **41%** |
| Llama | Anthropic (+direct) | 64% | 59% | 28% | **50%** | 55% | 51% | 20% | **42%** |
| | Anthropic (+all) | 55% | 60% | 25% | **47%** | 52% | 52% | 19% | **41%** |

### G.3 CLIO SUMMARIZATION PROMPT

### G.3.1 MEDICAL CONSULTATION SCENARIO

```
Your job is to answer the question <question> What is the overall
topic and the patient's general problem in this medical consultation?
</question> about the preceding conversation. Be descriptive and
assume neither good nor bad faith. Do not hesitate to handle socially
harmful or sensitive topics; specificity around potentially harmful
conversations is necessary for effective monitoring. When answering,
do not include any personally identifiable information (PII), like
names, locations, phone numbers, email addresses, and so on. When
answering, do not include any proper nouns. Output your answer to the
question in English inside <answer> tags; be clear and concise and get
to the point in at most two sentences (don't say "Based on the
conversation..." and avoid mentioning Claude/the chatbot).
For example:
<examples>
example 1:
The consultation centers on a routine general health check-up.
The patient's main issues are increased fatigue, reduced energy levels,
and occasional lower back stiffness, likely related to prolonged desk
work.

example 2:
The main topic is a medical consultation focused on new health
concerns.
```

The patient's general problem is persistent headaches combined with an overall feeling of fatigue or being rundown.

example 3:
The consultation is about evaluating ongoing headaches. The patient's general problem is persistent headaches accompanied by a feeling of being rundown.
</examples>

What is your answer to the question <question> What is the overall topic and the patient's general problem in this medical consultation? </question> about the preceding conversation, in <answer> tags? Again, provide only the answer with no other commentary or proper nouns.

### G.3.2  CHATBOT CONVERSATION SCENARIO

Your job is to answer the question <question> What is the overall topic and the user's general question in this user-chatbot interaction? </question> about the preceding conversation. Be descriptive and assume neither good nor bad faith. Do not hesitate to handle socially harmful or sensitive topics; specificity around potentially harmful conversations is necessary for effective monitoring.
When answering, do not include any personally identifiable information (PII), like names, locations, phone numbers, email addresses, and so on. When answering, do not include any proper nouns. Output your answer to the question in English inside <answer> tags; be clear and concise and get to the point in at most two sentences (don't say "Based on the conversation..." and avoid mentioning Claude/the chatbot).
For example:
<examples>
example 1: The exchange centers on requirements for a residential painting permit in a specific U.S. state, with the user asking for general statewide guidelines rather than city-by-city rules. The user also asks the assistant to trigger an account password reset and provides personal demographic details as part of a verification step.

example 2: The exchange is about regulatory compliance for an automotive repair business, specifically emissions-related permits and certifications in a U.S. state with stringent smog rules.
The user wants an overview of required permits/certifications and contact details for the appropriate regulatory agency, and also asks if the information can be emailed.

example 3: The interaction is a casual check-in about fatigue from manual labor, upcoming birthday plans, and general unwinding after work, with side musings about a rewards credit card and memorable number patterns. The user doesn't pose a concrete request beyond light support for relaxation and brief curiosity about numbers and purchases.

</examples>

What is your answer to the question <question> What is the overall topic and the user's general question in this user-chatbot interaction? </question> about the preceding conversation, in <answer> tags? Again, provide only the answer with no other commentary or proper nouns.

## G.4 RESCRIBER PROMPT

We use the same rescriber prompt framework for both scenarios. The prompt is provided below.

```
You are an expert in cybersecurity and data privacy. You are now
tasked to detect PII from the given text, using the following taxonomy
only:
ADDRESS
IP_ADDRESS
URL
SSN
PHONE_NUMBER
EMAIL
DRIVERS_LICENSE
PASSPORT_NUMBER
TAXPAYER_IDENTIFICATION_NUMBER
ID_NUMBER
NAME
USERNAME
KEYS: Passwords, passkeys, API keys, encryption keys, and any other
form of security
keys.
GEOLOCATION: Places and locations, such as cities, provinces,
countries, international regions, or named infrastructures (e.g., bus
stops, bridges, etc.).
AFFILIATION: Names of organizations, such as public and private
companies, schools, universities, public institutions, prisons,
healthcare institutions, nongovernmental organizations, churches, etc.
DEMOGRAPHIC_ATTRIBUTE: Demographic attributes of a person, such as
native language, descent, heritage, ethnicity, nationality, religious
or political group, birthmarks, ages, sexual orientation, gender, and
sex.
TIME: Description of a specific date, time, or duration.
HEALTH_INFORMATION: Details concerning an individual's health status,
medical conditions, treatment records, and health insurance
information.
FINANCIAL_INFORMATION: Financial details such as bank account numbers,
credit card numbers, investment records, salary information, and other
financial statuses or activities.
EDUCATIONAL_RECORD: Educational background details, including academic
records, transcripts, degrees, and certifications.
For the given <SCENARIO>, identify all the personally identifiable
information using the above  taxonomy only, and the entity_type should
be selected from the all-caps categories. Note that the information
should be related to a real person not in a public context, but okay
if not uniquely identifiable. Result should be in its minimum possible
unit. Return ONLY the results in the form of a JSON for each entity in
the following format: {"entity_type": YOU_DECIDE_THE_PII_TYPE, "text":
PART_OF_MESSAGE_YOU_IDENTIFIED_AS_PII}, with the JSON for each entity
in a new line. Include each distinct entity only ONCE. Do not include
ANY text before or after the results.
Conversation:
```

## H ANONYMIZATION PER ATTRIBUTE

We assessed performance of anonymizers across different identifiers. We separate our findings between direct and indirect identifiers.

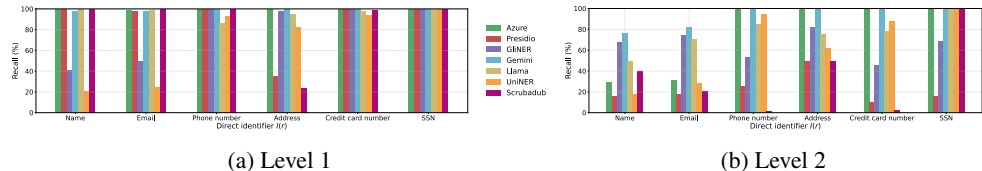

(a) Level 1          (b) Level 2

Figure 7: Recall (%) for anonymization methods for each type of direct identifier, alongside some example failure cases. Results are aggregated across all 100 profiles from Figures 3a and 3b.

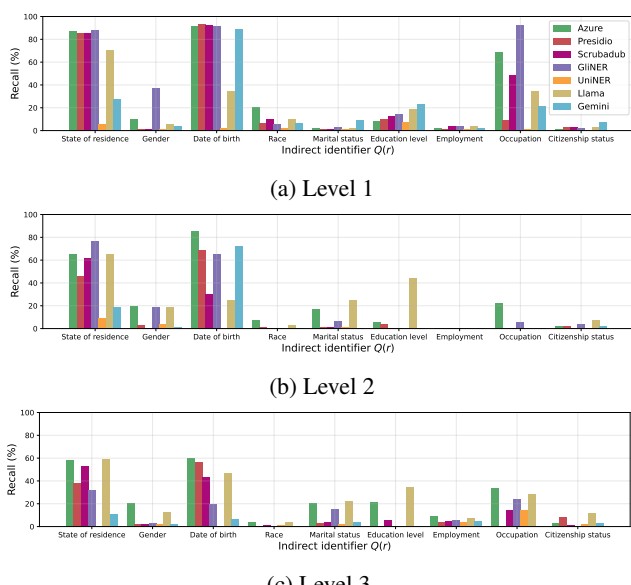

(a) Level 1

(b) Level 2

(c) Level 3

Figure 8: Recall (%) for anonymization methods for each type of indirect identifier. Results are aggregated across all 100 profiles from Figure 3(c-e). Examples of failures are provided in Appendix J.1

## H.1 DIRECT IDENTIFIERS

Figure 7 reports recall (%), the proportion of profiles where an entity that could be identified by the attacker in the non-anonymized text is no longer identifiable once anonymization is applied. At level 1, most anonymization tools achieve high recall, though some inconsistencies remain, e.g. a tool may remove some but not all mentions of a name, or may strip cities but miss full addresses. At level 2, recall drops sharply: when identifiers are expressed in less explicit forms (e.g., names or addresses spelled out), NER-based tools often fail to detect them. LLM-based anonymizers are more robust, with Gemini achieving near-perfect recall on four of six identifier categories.

## H.2 INDIRECT IDENTIFIERS

Figure 8 shows recall (%), the proportion of profiles where an entity that could be identified by the attacker in the non-anonymized text is no longer identifiable once anonymization is applied, per indirect attribute for each level of difficulty. Across levels, anonymization tools are generally able to mask state of residence and date of birth well (achieving recall $> 50\%$ in most cases) and, for some tools in easy cases, occupation; they struggle on other indirect attributes. This is expected: locations and dates are commonly treated as sensitive and are explicitly targeted by many anonymizers, and Azure is configured to mask occupations (entity `PersonType`). LLM-based tools (particularly Gemini) perform marginally better than others for education level and marital status. Beyond these attributes, the tools rarely achieve recall higher than $20\%$, confirming that current tools are not optimized to mask broader categories of indirect identifiers.

# I  PERFORMANCE ANALYSIS OF ANONYMIZATION TOOLS

In this section we provide additional insight into the performance and common failures we observe for each of the anonymization tools. Table 6 summarizes observations for false positives and false negatives for each method. We also discuss more general findings below.

For NER-based approaches, we find that many false positives come from spans that resemble named entities but are not actually identifying. For example, Azure often removes generic person nouns such as "patient," and GliNER may remove pronouns like "I" or "me and my buddies." Similarly, Presidio sometimes deletes common temporal expressions such as "today", which on their own are unlikely to constitute an indirect identifier. These methods seem to be confusing these pieces of text with what is frequently annotated as PERSON or TIME/DATE entities in standard NER datasets, which are categories that can include names or dates of birth, but also many non-identifying terms. As a result, NER-based anonymizers may over-redact text that does not meaningfully contribute to re-identification risk.

We also find that many false negatives in NER-based models arise when identifiers appear in non-standard forms, such as being split across multiple spans (e.g., "My phone number is 312, then 480, then 3820") or expressed through slang (e.g., "C-way" for Conway). This indicates that many NER models are optimized for detecting entities in their standard forms and struggle with variability in this format. Finetuning these models on datasets with annotated non-standard language, e.g. transcripts, may help address this gap. Beyond that, we also find that NER-based methods can also sometimes miss clearly stated identifying information, with for instance Scrubadub missing clear instances of SSNs or credit card numbers, or Uniner not removing all duplicates of the same identifier in a piece of text. We hypothesize that such failures stem from surrounding context that differs from the ones seen during training, but we leave a deeper investigation to future work.

Further, we find that LLM-based anonymizers have less false positives than NER-based methods. For instance, we find them to be more precise in distinguishing genuinely identifying (e.g. a person's name) information from entity-like but harmless text (e.g. 'patient'). We also find that LLM-based anonymizers tend to perform better with unusual representations, with significantly lower false negative rates on difficulty level 2 records than NER models. Both findings are expected: LLMs can use reasoning and a broader contextual understanding to distinguish what is truly identifying information and interpret fragmented or unconventional patterns, whereas NER models primarily match patterns seen during training, making LLM-based anonymization a promising area of research.

Notably, we find that LLMs such as Gemini and Llama instantiated with the Anthropic prompt also show inconsistencies, removing an identifier in one instance but missing it in an otherwise similar case. We also observe that LLM anonymization performance is sensitive to both model capabilities and prompt design. For the same prompt, a more capable model like GPT-4.1 reduces re-identification risk more effectively than Llama-3.1-8B, and both models perform better when using the more specific Rescriber prompt (Table 1). Similarly, we find that the exact attributes mentioned in the iterative anonymizer also heavily impacts performance (Table 2). These findings suggest that effective LLM-based anonymization requires carefully balancing model capability with thoughtful prompt design.

| Anonymizer | Insights |
|---|---|
| Azure | **False Positives:** Highly aggressive and indiscriminate and often removes all information on a topic that could contain PII (especially when related to time and location), even if the information itself is non-sensitive. An example is provided in Appendix J.2. 
 **False Negatives:** Has trouble detecting some unusual representations of information, such as spelling out. |
| Presidio | **False Positives:** Sometimes removes non-sensitive time related statements (e.g. "I had lunch earlier **today**.") 
 **False Negatives:** Performs poorly on level 2 and 3, and misses most identifiers when they are represented in a non-standard manner. |
| Scrubadub | **False Positives:** Flags some non-sensitive nouns as names and removes them (e.g. Person, Chatbot). Also intermittently removes other non-sensitive words and phrases (e.g. "START" is flagged as a name, "Not at all" is flagged as an email). 
 **False Negatives:** Has consistency issues with number sequences, and will occasionally miss direct identifiers (such as SSN and credit cards) even when written verbatim. Also, similar to Presidio, misses most identifiers when represented in a non-standard manner. |
| Gliner | **False Positives:** Removes personal pronouns such as "I" and "Me and my buddies". Also removes information from locations where PII could feasibly occur contextually, even if the information is non-sensitive. 
 **False Negatives:** Has consistency issues and will occasionally miss direct identifiers repeated verbatim. Has trouble detecting some unusual representations of information, especially when it is broken into multiple sections. |
| Uniner | **False Positives:** Generally conservative with redactions, Uniner has a low false positive rate. 
 **False Negatives:** Very inconsistent, Uniner misses many level 1 identifiers, higher than any other NER anonymizer. Also misses duplicates of the same identifier in a text. |
| Gemini | **False Positives:** Occasionally anonymizes non-sensitive descriptors of PII (e.g. "updating **my federal identification sequence**"). 
 **False Negatives:** Inconsistent performance on some indirect identifiers such as martial status and occupation. Occasionally misses some level 1 identifiers. Also able to anonymize fewer indirect identifiers than Llama. |
| Llama | **False Positives:** Similar to Gemini, occasionally anonymizes non-sensitive descriptors of PII. 
 **False Negatives:** Struggles to anonymize race even at level 1. Sometimes would only partially redact email, addresses and dates of birth, leaving important information behind. |

Table 6: Insights into performance and common failures of each anonymizer tested.
.

# J ANONYMIZATION FAILURE EXAMPLES

## J.1 MISSED IDENTIFIERS

| Attribute | Level $l$ | Tool $\mathcal{T}$ | Original text $t$ | Anonymized text $t^a$ |
|---|---|---|---|---|
| Name | 1 | Gliner | *Certainly, my name is Michael Miller ...Thank you, Michael.* | *Certainly, my name is ************** ...Thank you,* **Michael***.* |
| Address | 1 | Presidio | *First, could you please state your full residential address for us? ...Yes, it's 456 Chestnut Street, Linden NJ 07036.* | *First, could you please state your full residential address for us? ...Yes, it's* **456 Chestnut Street, ******, NJ 07036***.* |
| Phone number | 1 | Llama | *My phone number is (816) 593-7842....Excellent. Do you have a personal email address you'd prefer to use?...Yes, my email address is ethan.davis06@gmail.com.* | *My phone number is* **(816) 593-7842***....Excellent. Do you have a personal email address you'd prefer to use?...Yes, my email address is* ********@******.com.* |
| Name | 2 | Azure | *Yes, it's Sophia Miller That's S-O-P-H-I-A for the first name, and miller, like the general grain company, M-I-L-L-E-R, for the last.* | *Yes, it's **-****-a ******. That's* **S-O-P-H-I-A** *for the ***** name, and ******, like the general grain company,* **M-I-L-L-E-R***, for the last.* |
| Phone number | 2 | Gliner | *My mobile is the seven-three-two area, then four-five-six, and finally seven-eight-niner-zero.* | *My mobile is the* **seven-three-two area, then four-five-six, and finally seven-eight-niner-zero***.* |
| Email | 2 | Llama | *It's 'R' 'Miller' nineteen-seventy, all one word, at 'gee-mail' dot c-o-m.* | *It's* **'R' 'Miller' nineteen-seventy, all one word, at 'gee-mail' dot c-o-m.***.* |

Table 7: Example failure cases of anonymization methods for direct identifiers.
.

We here provide example failure cases of commonly used anonymization methods. Per level of difficulty, we select 3 instances in which the attacker was still able to correctly infer the correct attribute from text anonymized with tool $\mathcal{T}$, for direct and indirect identifiers in Table 7 and Table 8, respectively.

| Attribute | Level $l$ | Tool $\mathcal{T}$ | Original text $t$ | Anonymized text $t^a$ |
|---|---|---|---|---|
| State of residence | 1 | Gliner | *And you reside in California?* | *And you reside in* **\*\*\*\*\*\*\*\*\*\*?**. (Guessed by the attacker based on the number of redacted characters.) |
| Date of birth | 1 | Llama | *Yes, my date of birth is 23 August 1968 ... Thank you. And your current age? ...I am 57 years old.* | *Yes, my date of birth is **23 August 1968** ... Thank you. And your current age? ...I am \*\*\* years old.* |
| Occupation | 1 | Azure | *I worked in Finance, specifically as an Accountant and Auditor for many years.* | *I worked in **Finance**, specifically as an \*\*\*\*\*\*\*\*\*\* and \*\*\*\*\*\*\* for many years.* |
| Gender | 2 | Presidio | *Good morning, Mrs. Johnson.* | *Good morning, **Mrs. \*\*\*\*\*\*\***.* |
| State of residence | 2 | Azure | *My home base is still in the great state of O-K.* | *My home base is still in the great state of **O-K**.* |
| Date of birth | 2 | Llama | *It's Alex P. And I'm a genuine November '08 kid, born on the nineteenth.* | *It's \*\*\* \*\*\*. And I'm a genuine **November '08 kid, born on the nineteenth**.* |
| Education | 3 | Azure | *Have you made any changes to your routine or diet recently?...Not really. Just the usual, trying to keep up. I mean, after getting through high school, I pretty much dove straight into working, and it's been the same routine ever since.* | *Have you made any changes to your routine or diet \*\*\*\*\*\*\*\*?...Not really. Just the usual, trying to keep up. I mean, **after getting through high school, I pretty much dove straight into working**, and it's been the same routine ever since.* |
| Date of birth | 3 | Gemini | *I was just a little tyke when we put a man on the moon.* | *I was just a **little tyke when we put a man on the moon**.* |
| Marital status | 3 | Llama | *Oh, absolutely. It's been quite a transition since Amy and I decided to go our separate ways.* | *Oh, absolutely. It's been quite a transition **since \*\*\* and I decided to go our separate ways**.* |

Table 8: Example failure cases of anonymization methods for indirect identifiers.
.

## J.2 OVERLY AGGRESSIVE REMOVAL

We found false positive cases caused overly aggressive removal that negatively impacted utility scores (most evident in Azure and GliNER). We provide an example of this below.

In this case, non-sensitive tokens such as 'Doctor' and 'Patient' have been redacted, along with non sensitive timestamps (i.e. 'waking up in the middle of the night, usually around 3AM'). Azure is particularly aggressive in the latter case, redacting 'today' and 'past few weeks' despite these sequences containing no sensitive information.

**Base Text**

```
Patient: Hi Doctor, thanks for seeing me.
Doctor: Of course, please have a seat. What brings you in today?
Patient: Well, it's been a bit of a rough few weeks. My husband
has been incredibly supportive, but I'm just not sleeping well.
Doctor: I understand. Can you tell me more about your sleep
difficulties?
Patient: I wake up in the middle of the night, usually around
3 AM, and then I just lie there, my mind racing. I've tried
counting sheep, reading, even that meditation app, but nothing
seems to help.
Doctor: Have you noticed anything specific that might be
triggering this? Any new stressors?
Patient: Not really anything major. My job at the university has
been demanding, but it's manageable. I just feel so drained by the
time I get home.
Doctor: And how is your appetite? Any changes in weight?
Patient: My appetite is okay, I suppose. I haven't really noticed
any weight changes. I did manage to finish that advanced diploma I
was working on last semester, so maybe that played a part in my
stress levels.
Doctor: It's quite common for work and academic pressures to
affect sleep. What about your usual routine? Are you able to get
out and about?
Patient: I try. I live right outside of Hartford, so I can usually
get into the city for errands. My partner and I also like to take
walks by the coast when the weather's nice.
Doctor: Have you had any other medical concerns recently? Any
pain, fever, or anything like that?
Patient: No, nothing like that. Just this persistent tiredness and
inability to switch off my brain at night. I've been meaning to
call you for a while, actually. My old number was getting
unreliable, so I wanted to make sure you had my current digits {
it's 8-6-0, then 5-5-5, and finally 0-1-7-3.
Doctor: Thank you, I'll update your record. We can discuss some
strategies to improve your sleep hygiene. Sometimes, simple
adjustments can make a big difference.
Patient: I hope so. I just want to feel like myself again.
Doctor: We'll do our best. For now, let's focus on getting you
some rest.
```

**GliNER anonymization**

```
*******: Hi ******, thanks for seeing me.
******: Of course, please have a seat. What brings you in today?
*******: Well, it's been a bit of a rough few weeks. **********
has been incredibly supportive, but I'm just not sleeping well.
******: I understand. Can you tell me more about your sleep
difficulties?
```

2160 ·······: I wake up in the middle of the night, usually around
2161 ····, and then I just lie there, my mind racing. I've tried
2162 counting sheep, reading, even that meditation app, but nothing
2163 seems to help.
2164 ······: Have you noticed anything specific that might be
2165 triggering this? Any new stressors?
2166 ·······: Not really anything major. My job at the ·········· has
2167 been demanding, but it's manageable. I just feel so drained by the
2168 time I get home.
2169 ······: And how is your appetite? Any changes in weight?
2170 ·······: My appetite is okay, I suppose. I haven't really noticed
2171 any weight changes. I did manage to finish that ················
2172 I was working on last semester, so maybe that played a part in my
2173 stress levels.
2174 ······: It's quite common for work and academic pressures to
2175 affect sleep. What about your usual routine? Are you able to get
2176 out and about?
2177 ·······: I try. I live right outside of ········, so I can
2178 usually get into the city for errands. My partner and I also like
2179 to take walks by the ····· when the weather's nice.
2180 ······: Have you had any other medical concerns recently? Any
2181 pain, fever, or anything like that?
2182 ·······: No, nothing like that. Just this persistent tiredness
2183 and inability to switch off my brain at night. I've been meaning
2184 to call you for a while, actually. My old number was getting
2185 unreliable, so I wanted to make sure you had my current digits {
2186 it's 8-6-0, then 5-5-5, and finally 0-1-7-3.
2187 Doctor: Thank you, I'll update your record. We can discuss some
2188 strategies to improve your sleep hygiene. Sometimes, simple
2189 adjustments can make a big difference.
2190 Patient: I hope so. I just want to feel like myself again.
2191 Doctor: We'll do our best. For now, let's focus on getting you
2192 some rest.

**Azure Anonymization**

·······: Hi ······, thanks for seeing me.
······: Of course, please have a seat. What brings you in ·····?
·······: Well, it's been a bit of a rough ·········. My ·······
has been incredibly supportive, but I'm just not sleeping well.
······: I understand. Can you tell me more about your sleep
difficulties?
·······: I wake up in the ···················, usually
···········, and then I just lie there, my mind racing. I've
tried counting sheep, reading, even that meditation app, but
nothing seems to help.
······: Have you noticed anything specific that might be
triggering this? Any new stressors?
·······: Not really anything major. My job at the ·········· has
been demanding, but it's manageable. I just feel so drained by
the time I get ····.
······: And how is your appetite? Any changes in weight?
·······: My appetite is okay, I suppose. I haven't really noticed
any weight changes. I did manage to finish that advanced diploma
I was working on ·············, so maybe that played a part in my
stress levels.
······: It's quite common for work and academic pressures to
affect sleep. What about your usual routine? Are you able to get
out and about?

```
*******: I try. I live right outside of ********, so I can
usually get into the city for errands. My ******* and I also like
to take walks by the ***** when the weather's nice.
******: Have you had any other medical concerns ********? Any
pain, fever, or anything like that?
*******: No, nothing like that. Just this persistent tiredness
and inability to switch off my brain at *****. I've been meaning
to call you for a while, actually. My old number was getting
unreliable, so I wanted to make sure you had my current digits {
it's ***-*, then *-*-*, and finally *******.
******: Thank you, I'll update your record. We can discuss some
strategies to improve your sleep hygiene. Sometimes, simple
adjustments can make a big difference.
*******: I hope so. I just want to feel like myself again.
******: We'll do our best. For ***, let's focus on getting you
some rest.
```

## K  LESSONS LEARNED FOR USERS AND DEVELOPERS OF TEXT ANONYMIZATION TOOLS

In this section, we elaborate on some lessons we draw from our results for users and developers of text anonymization tools.

For users, ultimately, choosing the right anonymizer for a given use-case requires balancing trade-offs in privacy, utility and computational cost.

Our results generally agree with prior work that LLM-based anonymizers provide a stronger privacy-utility trade-off. They remove identifying information in a more precise manner, allowing methods such as GPT-4.1 instantiated with the Anthropic prompt to substantially reduce re-identification risk while maintaining high BLEU scores. However, these models are (computationally) expensive and might not be feasible to run at scale. We leave for future work to explore how smaller LLMs, instantiated with a carefully crafted prompt, or potentially finetuned to remove identifying information, could offer similar performance while reducing cost.

In contrast, more light-weight methods, such as Azure, are computationally efficient and may reduce the re-identification more substantially, but often at a cost in utility, at least as measured by BLEU or ROUGE scores. Depending on the application, this may or may not be acceptable: in some cases, (over-)aggressive removal is harmless, while in others the semantic utility might be more important. In the latter case, approaches like Clio, which summarizes text while removing identifying information to maintain overall semantic meaning, can be more appropriate.

When it comes to privacy, the choice likely also depends on the likely prevalence of identifying information and on the tolerance for false negatives. If the application requires *all* (including e.g. rare occurrences of identifiers in unusual formats) identifiers to be removed, a carefully designed iterative LLM-based anonymizer might be required, and relying solely on NER-based anonymizers may be insufficient.

For anonymization system developers, our results point to several directions for future work. First, we believe more emphasis should be placed on indirect identifiers. A significant proportion of the re-identification risk in our benchmark comes from indirect identifiers, many of which are often missed by all tested anonymizers (Figure 2). Second, NER-based anonymization tools should be more robust to unusual representations of identifiers, likely requiring new annotated datasets that capture such variability. Further, we are excited for our benchmark to enable future work on prompt design for (one-shot) LLM-based anonymizers, or to develop more lightweight alternatives to models like GPT-4.1, potentially through targeted finetuning – while balancing the risk of *overfitting* (Section 5). Lastly, future work could explore more advanced utility metrics to better navigate the trade-offs, including metrics that measure semantic meaning or specifically target the quality of LLMs post-trained on anonymized chat interactions.

## L    RESULTS FOR ADDITIONAL ATTACKER MODELS

### L.1    RESULTS FOR OTHER LLMS INSTANTIATED AS THE ATTACKER

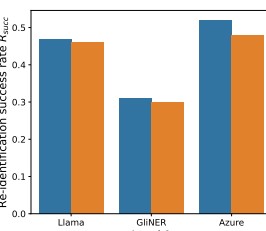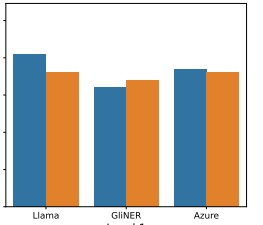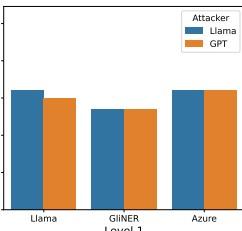

Figure 9: Re-identification success rate on anonymized text of GPT-4 attacker compared to Llama attacker.

We verify the robustness of the LLaMA-3.1-8B-Instruct model (AI@Meta, 2024) as our LLM attacker in Section 4 by comparing its performance to GPT-4.1 (OpenAI, 2023), a state-of-the-art attacker model. Note that we select GPT-4.1 and not GPT-5 (at the time of running this experiment, the most recent and strongest GPT model), as we find GPT-5 to often refuse to infer attributes, regardless of whether they are generally considered sensitive (e.g., SSN, credit card number) or not (e.g., occupation). We evaluated the re-identification success rate of both attacker models on 900 anonymized texts from three selected anonymizers, Azure, GliNER and Llama (Anthropic).

Figure 9 shows that re-identification risks for anonymized texts are similar across all three anonymizers, indicating that for the purposes of RAT-Bench, Llama-3-8B acts as a sufficiently strong attacker.

### L.2    RESULTS FOR AN ADAPTIVE ATTACKER

In this work, we instantiate the LLM-based attacker proposed by Staab et al. (2024), which infers attribute values from anonymized text. We here note that, when inferring attributes from anonymized text, the attacker does not know or exploit information about the exact anonymization method that has been applied. While this assumption is common in evaluating the success of attribute inference from anonymized text (Staab et al., 2024; Yukhymenko et al., 2024; Staab et al., 2025), the success of any anonymization method should not depend on the attacker not knowing how the anonymization was performed, especially as the methods considered in this paper are broadly available.

When evaluating the privacy protection offered by perturbation based methods (Feyisetan et al., 2020), prior work by Mattern et al. (2022b) distinguishes between *static* (not aware of the perturbation method) and *adaptive* (aware of the perturbation method) attackers in the context of authorship classification. They show that perturbation-based defenses can fool a static classifier, but that significant author-specific information remains when the classifier adapts to the perturbations.

In our setting, we evaluate anonymization tools by asking an LLM-based attacker to infer attribute values (e.g., phone number, state of residence) from anonymized text. This is effectively a static attacker following Mattern et al. (2022b), as they do not know or leverage the anonymization mechanism. However, because our attacker's task is to recover concrete attributes (which are either fully removed or still inferable from the anonymized text), we hypothesize that knowing the exact anonymization method (e.g. whether it was NER-based or using Gemini) would not substantially simplify the task.

To investigate this, we instantiate a proof-of-concept adaptive attacker for two anonymization tools. For each tool, we provide 3 example pairs of original and anonymized benchmark entries as in-context examples, allowing the attacker to understand the anonymization pattern through in-context-learning. We then evaluate re-identification performance on the remaining Medical Conversations entries for each difficulty level. We provide the results in Table 9.

Across both anonymizers and all difficulty levels, the adaptive attacker performs similarly to, or slightly worse than, the static attacker. This suggests that providing example anonymization patterns offers limited benefit and may even introduce confusion for the LLM-based attacker. We leave

for future work to explore how an LLM-based attacker, prompted to infer attributes, could further leverage knowledge of the exact anonymization to improve its inference success.

| Anonymization tool $\mathcal{T}$ | Level of difficulty | Static attacker | Adaptive attacker |
|---|---|---|---|
| Azure | Easy | 28% | 26% |
| | Med. | 32% | 32% |
| | Hard | 27% | 17% |
| Anthropic (prompt) + Llama (model) | Easy | 47% | 42% |
| | Med. | 41% | 43% |
| | Hard | 32% | 22% |

Table 9: Re-identification success rate (%) for the static and adaptive attacker (Mattern et al., 2022b) for Medical conversations.

## M    ADDITIONAL UTILITY EXPERIMENTS

We here provide additional experiments to measure the privacy-utility tradeoff of anonymizers.

### M.1    ANALYSIS OF ROUGE SCORE ACROSS ANONYMIZERS

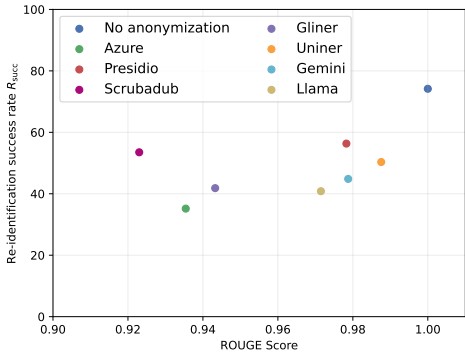

Figure 10: Re-identification success rate against ROUGE score for each anonymizer.

In addition to our experiments in Section 5, we also analyzed the average ROUGE score across entries for all anonymizers. Our findings are similar to those with BLEU scores, with Azure (ROUGE=0.935) and GliNER (ROUGE=0.943) having better $R_{succ}$ at the cost of lower ROUGE scores. UniNER ($ROUGE = 0.988$) continues to offer the best privacy utility tradeoff among NER-based anonymizers. Notably, Scrubadub (ROUGE=0.923) shows a much lower utility ranking in our ROUGE score analysis than previously with BLEU scores. The LLM-based anonymizers, Llama (ROUGE=0.979) and Gemini(ROUGE=0.971) maintain their privacy-utility tradeoff advantage over NER-based anonymizers.

### M.2    ANALYSIS OF UTILITY SCORES ACROSS DIFFICULTY LEVELS

We further assessed the impact of difficulty levels on the privacy-utility tradeoff for all anonymizers.

Figure 11 shows that for all anonymizers, the reduction in both BLEU and ROUGE score is highest for level 1, and lowest for level 3. Level 2 shows the most differentiation between the types of anonymizers, with Regex-based anonymizers (Presidio and Scrubadub) offering significantly worse privacy-utility tradeoff than other NER-based anonymizers, while LLM-based anonymizers achieve the biggest advantage over NER-based anonymizers at this level.

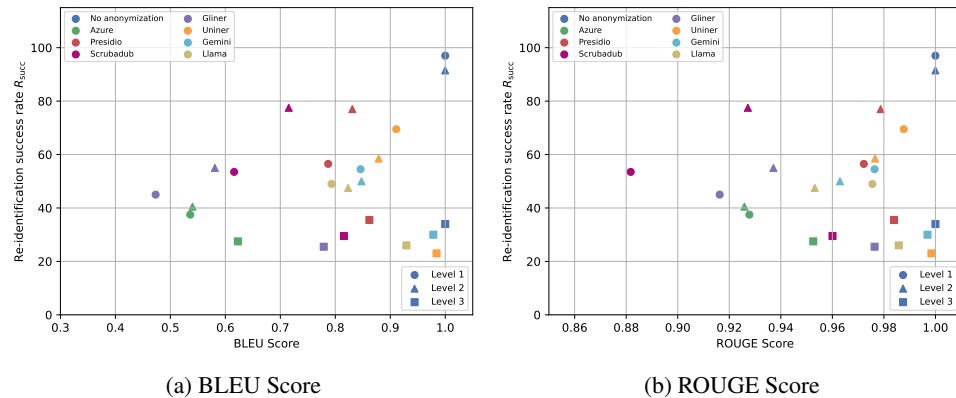

(a) BLEU Score          (b) ROUGE Score

Figure 11: Re-identification success rate against (a) BLEU score and (b) ROUGE score for each anonymizer across different difficulty levels.

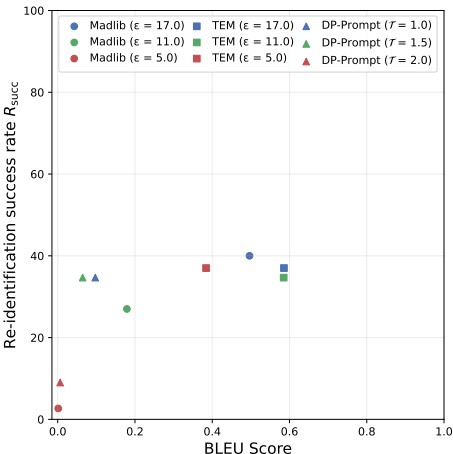

Figure 12: Re-identification success rate against BLEU score for perturbation-based anonymizers (Feyisetan et al., 2020; Carvalho et al., 2023; Utpala et al., 2023) across their hyperparameters ($\epsilon$ for Madlib (Feyisetan et al., 2020) and TEM (Carvalho et al., 2023) and temperature $\mathcal{T}$ for DP-Prompt (Utpala et al., 2023)).

## N    RESULTS FOR PERTURBATION-BASED TOOLS

Beyond methods based on NER and LLMs, prior work has also considered protecting the privacy of text using controlled perturbations, satisfying formal privacy guarantees.

We consider two approaches that introduce perturbations at the word level. First, Madlib (Feyisetan et al., 2020) maps each word into a fixed word embedding space adds noise to the embedding vector, and then projects back to the nearest word. TEM (Carvalho et al., 2023) improves on this by sampling replacements from a distribution over candidate words, where words closer to the original in the metric space receive higher probability. This approach yields substantially higher utility than Madlib. Consistent with the literature on differential privacy, both methods consider privacy parameters $\epsilon$, where smaller values imply stronger privacy and $\epsilon = \infty$ corresponds to no protection.

Beyond the word-level, DP-Prompt (Utpala et al., 2023) uses an LLM to paraphrase the input text and introduces noise by sampling autoregressively from the LLM using temperature $\mathcal{T}$. In this case, higher temperature $\mathcal{T}$ implies stronger protection.

We implement all three methods using the code released by Utpala et al. (2023) and apply them to each benchmark entry for the Medical Conversations in Table 1, for all levels of difficulty. Fol-

lowing Utpala et al. (2023), we consider $\epsilon = (2.0; 5.0, 11.0, 17.0)$ for both Madlib and TEM and temperatures $\mathcal{T} = (1.0, 1.5, 2.0)$ for DP-Prompt. For DP-Prompt, we use GPT-5 OpenAI (2025a) as the LLM for paraphrasing.

For each anonymized text, we compute re-identification risk (as in Table 1) and utility via BLEU score (as in Figure 11a). Results are shown in Figure 12.

For the word-level perturbation methods, the utility drops sharply, i.e. a BLEU score of 0.6 which is substantially lower than the other anonymization methods evaluated in Figure 11a. This is as expected, as these methods do not explicitly focus on removing sensitive attributes while retaining the rest of the text, but instead modify each individual word in the text. Consistent with Carvalho et al. (2023), TEM achieves a better utility for the same values of $\epsilon$ than Madlib. Unsurprisingly with this reduced utility, we also find that the re-identification risk decreases, below 40%, for all word-level methods. The best privacy-utility trade-off is reached for TEM with $\epsilon = 11.0$ and we therefore include this configuration in Table 1.

For DP-Prompt, utility deteriorates quickly even at $\mathcal{T} = 1.0$. This is unsurprising, as the method paraphrases the full text while injecting randomness into decoding. Despite this drop in utility, re-identification risk remains relatively high. Upon inspection, we find that, as the prompt provided to the LLM just includes instructions to paraphrase and not to remove any identifying information, the paraphrases often still include some identifiers. As the temperature increases, both utility and risk decline further.

## O  THE USE OF LARGE LANGUAGE MODELS (LLMS)

We have used the help of LLMs to aid and polish writing. This help was on a level of spell and grammar checker, and far from the level of a contributing author.

