# OpenReview forum: "Unique between the lines: benchmarking re-identification risk for text anonymization"
_ICLR.cc/2026/Conference — Submitted to ICLR 2026_

### Official Review · Reviewer_8xbg · 2025-10-21

**Soundness:** 2
**Presentation:** 2
**Contribution:** 1
**Rating:** 2
**Confidence:** 5

**Summary:**

This paper introduces a benchmark on individual reidentification risks from natural texts, aimed at evaluating anonymization methods in the face of an LLM adversary. The benchmark is synthetically constructed by sampling US demographic tabular data and synthesizing text around it using an LLM. The synthesized text may contain both indirect identifiers (e.g., sex) and direct identifiers (e.g., social security number). There are three difficulty levels defined, easy, medium, and hard, depending on the types of identifiers that are contained and ont their difficulty of extraction. Using their benchmark, the authors evaluate several anonymization tools, and conclude that they all still pose a large risk of reidentification post-anonymization.

**Strengths:**

- Important problem.
- Linking the inferred attributes to the concrete reidentification risk is a useful metric.

**Weaknesses:**

Overall, the paper lacks novelty.
- The introduced benchmark does not offer sufficient addition over existing datasets for this purpose, e.g., SynthPAI (Yukhymenko et al. 2024). Also, it is unclear why a reconstruction of such a dataset is even needed, why could one not just extend SynthPAI to include direct identifiers.
- It is clear that direct identifiers, if left in the text, will lead to reidentification. As such, *any* prior benchmark of classical anonymization, NER, etc. has already measured this risk. This is not a novel contribution.
- Also, the reidentification risk from direct identifiers, is simply a bijective and monotonic function of the inferred attributes. As such, already the results of prior work on LLM inference against anonymized text (e.g., Staab et al. 2024 & 2025) already carry exactly the same conclusions. Simply a conversion can be applied to the attribute inference accuracy metrics. And this conversion is not a contribution of this paper either, as it was introduced, according to the authors' citations as well, by Rocher et al. (2019).
- The paper lacks novel experiments and conclusions. The difficulty level-based separations in observed inference patterns have already been observed by Staab et al. (2024) & (2025). The poor utility of NER-based classical anonymmization tools has also been already observed by Staab et al. 2025. The fact that NER-based anonymizers do better with direct identifiers/direct inclusions of the indirect identifiers and struggle with contextual clues has been observed, once again, by Staab et al. (2024) & (2025). The cost of LLM-anonymization has also already been analyzed by Staab et al. (2025). The sole truly novel experiment is included in Figure 3, showing a growing reidentification risk with an increased number of attributes in the text. Note however that this experiment's results are unfortunately not surprising, given the overall monotonicity of reidentification.

Further, the paper is not state-of-the-art.
- The LLM adversary is a weak local model; understimating the privacy risks. Other works on LLM-based inference risks argue for strong adversaries for this reason (Staab et al. 2025).
- The LLM-based anonymization method used in the evaluation is, in the face of existing literature to this end, ad-hoc. The authors should have evaluated the LLM-based anonymization methods that they cite themselves as well in Section 2.
- The introduced benchmark dataset is not as realistic, robust, and versatile as datasets used in other works. Unclear why these were not employed, or why their synthetic construction quality criteria were not observed (e.g., Yukhymenko et al. 2024).

Minor.
- The figures on page 8 are rather small and difficult to read.
- The x-axis in Figure 1 (a) is unclear. How can one have a monotonous function (ordered) over these categories (independent)?

**References**

L Rocher et al. Estimating the success of re-identifications in incomplete datasets using generative models. Nature Communications 2019.

R Staab et al. Beyond memorization: Violating privacy via inference with large language models. ICLR 2024.

H Yukhymenko et al. A Synthetic Dataset for Personal Attribute Inference. NeurIPS 2024.

R Staab et al. Large language models are advanced anonymizers. ICLR 2025.

**Questions:**

See weaknesses.

---

> ### Author Response · Authors · 2025-11-21
> **Response to Reviewer 8xbg**
>
> We thank the reviewer for their time spent on the paper and their thoughtful feedback. Below, we address each of the points raised.
>
> > Overall, the paper lacks novelty.
>
> While we greatly appreciate previous work in the literature, including [1,2,3], we respectfully disagree with the reviewer that our paper lacks novelty. We here proposes the first *end-to-end pipeline* that allows for *controlled leakage* of *direct and real-world indirect* attributes and the computation of *re-identification risk*. As we discuss below, all of these components are necessary to benchmark anonymization tools according to a realistic re-identification risk, in line with legal standards.
>
> We agree that the papers highlighted by the reviewer, Staab et al., 2024 [1], Yukhymenko et al., 2024 [2], and Staab et al., 2025 [3], represent important progress on attribute inference and LLM-based anonymization, and we already discuss and cite them in our paper:
> - Staab et al. (2024) show that LLMs can infer eight personal attributes from real Reddit profiles and that off-the-shelf anonymizers, namely Azure, reduce, but do not eliminate, inference accuracy.
> - Yukhymenko et al. (2024) introduce SynthPAI, a synthetic Reddit-style dataset where LLM agents generate posts conditioned on profiles containing the corresponding set of indirect attributes.
> - Building on these datasets, Staab et al. (2025) propose an LLM-based anonymization method that iteratively rewrites text until an LLM attacker can no longer successfully infer any attributes.
>
> These are part of a broader literature which we also build on and cite, including (i) work on the ambiguity and context-dependence of PII in text (Brown et al., 2022 [4]); (ii) LLM-based attribute inference and text anonymization (Liu et al., 2025 [5], Patsakis & Lykousas, 2023 [6]); and (iii) dedicated benchmarks and risk-based evaluations for text anonymization (Pilán et al., 2022; Manzanares-Salor et al., 2024; Singh & Narayanan, 2025).
>
> We consider our paper to be strongly complementary to previous work:
> - *Re-identification risk.* All prior work reports attribute-level inference accuracy, which, however, does not directly reflect re-identification risk. Different attributes vary in difficulty and importance, e.g. consistently inferring gender contributes to re-identification in different ways than other attributes and depend heavily on the context. Moreover, multiple attributes can co-occur in a single text, potentially compounding the risk. These are important considerations that accuracy alone fails to capture. Instead, we map correctly inferred identifiers to re-identification risk. Our approach incorporates both direct and indirect identifiers, each of which contributes differently and must be considered from a legal perspective, and systematically varies their co-occurrence within a single text.
> - *Grounded in real-world demographics.* When only indirect identifiers are correctly inferred, we use Rocher et al.’s (2019) [8] estimator to compute the re-identification risk across the entire US population. Incorporating real-world sets of indirect identifiers in benchmark construction, rather than synthetic ones, is essential for producing realistic, population-level estimates of re-identification risk.
> - *Systematic control.* We systematically vary the number, type and difficulty of direct or indirect identifiers inserted into each generated text. This enables rigorous benchmarking of a wide range of anonymization techniques, measuring their progress and evaluating their failure modes. Our benchmark methodology is furthermore easily extendable to other indirect identifiers, demographics, languages etc.
> - Thanks to the reviewer's suggestions, we also now include a methodology to evaluate the effectiveness of *iterative and newly proposed LLM anonymizers* across a range of tasks while avoiding overfitting (Section 5).

---

> ### Author Response · Authors · 2025-11-21
>
> We also address the reviewer’s points individually:
>
> > The introduced benchmark does not offer sufficient addition over existing datasets for this purpose, e.g., SynthPAI (Yukhymenko et al. 2024). Also, it is unclear why a reconstruction of such a dataset is even needed, why could one not just extend SynthPAI to include direct identifiers
>
> While SynthPAI makes an important contribution by generating synthetic Reddit-style data with ground-truth indirect attributes for evaluating inference accuracy, these attributes are entirely synthetic. Therefore, there is no well-defined “universe” over which to measure how unique a combination of inferred attributes is. For example, learning that a person is “80-89, female, income >10M ” or “30-39, female, income >100K ” may correspond to very different re-identification risks depending on the actual demographic distribution, which SynthPAI cannot model.
>
> Our benchmark addresses this by grounding attribute combinations in real U.S. population demographics, enabling meaningful estimates of re-identification risk. Although SynthPAI could be extended to include direct identifiers, this alone would not make it suitable for a holistic and realistic risk estimation.
>
> Also, while SynthPAI focuses on Reddit-style comments, we consider two different scenarios: medical consultations and AI-chatbot interactions - an increasingly relevant domain, especially as AI companies use user-chat data for model improvement [9,10,11] and, just last week, a court required OpenAI to release anonymized chat transcripts [12].
>
>
> > It is clear that direct identifiers, if left in the text, will lead to reidentification. As such, any prior benchmark of classical anonymization, NER, etc. has already measured this risk. This is not a novel contribution.
>
> There indeed exists interesting literature of NER benchmarks [7,14], which typically evaluate the detection of a fixed taxonomy of entities (e.g. names, locations, time expressions), *even though their privacy implications vary widely by context*. For instance, failing to mask a patient’s name in a medical transcript is a severe privacy breach, whereas masking generic nouns such as “patient” or vague time expressions such as “earlier today” (as we observe for tools such as Azure) may unnecessarily reduce utility without improving privacy (see our discussion in Appendix I). While we agree that prior work has shown that NER-based anonymizers are imperfect and may miss occurrences of direct identifiers [1,2,3,13], they do not provide an assessment of re-identification risk in anonymized text accounting for both direct and indirect identifiers (potentially co-occurring) .
>
> Our benchmark aims to address these limitations by injecting *5 distinct direct identifiers in a systemically controlled manner, i.e. varying their number and difficulty across scenarios*. This allows us to quantify the residual re-identification risk but also to discover new anonymizer failure cases (see Appendix I).
>
> > Also, the reidentification risk from direct identifiers, is simply a bijective and monotonic function of the inferred attributes. As such, already the results of prior work on LLM inference against anonymized text (e.g., Staab et al. 2024 & 2025) already carry exactly the same conclusions. Simply a conversion can be applied to the attribute inference accuracy metrics. And this conversion is not a contribution of this paper either, as it was introduced, according to the authors' citations as well, by Rocher et al. (2019).
>
> We assume here that the reviewer refers to re-identification risk from indirect identifiers. If so, we clarify how we compute the risk for anonymized text (Algorithm 2):
> - If the attacker correctly infers any direct identifier, the risk is set to 1.
> - Otherwise, we compute the risk based on the set of correctly inferred indirect identifiers using the method from Rocher et al. (2019). In this case, the risk indeed monotonically increases with each correctly inferred indirect identifier.
>
> Importantly, this risk not only depends on the type of attribute (e.g. state of residence), but also on *how unique the actual attribute value (and their combinations) is in the entire population* (e.g. correctly inferring that someone is from California incurs less risk than doing so for the state of Rhode Island). To our understanding, Staab et al. 2024 & 2025 only consider indirect identifiers, and report only inference accuracy, regardless of the attribute values or their co-occurrence, which does not directly imply re-identification. The proper computation of re-identification risk can not be easily applied post-hoc to other dataset, but needs to be built in by design in the benchmark.

---

> ### Author Response · Authors · 2025-11-21
>
> > The paper lacks novel experiments and conclusions. The difficulty level-based separations in observed inference patterns have already been observed by Staab et al. (2024) & (2025).  The poor utility of NER-based classical anonymmization tools has also been already observed by Staab et al. 2025. The fact that NER-based anonymizers do better with direct identifiers/direct inclusions of the indirect identifiers and struggle with contextual clues has been observed, once again, by Staab et al. (2024) & (2025). The cost of LLM-anonymization has also already been analyzed by Staab et al. (2025). The sole truly novel experiment is included in Figure 3, showing a growing reidentification risk with an increased number of attributes in the text.
>
> While we acknowledge that Staab et al. (2024,2025) observed that anonymization methods, primarily Azure, perform differently at different levels of difficulty, our approach is specifically designed to benchmark many anonymization methods across levels, identify failure cases, and track progress. We now further elaborate on method-specific failure cases in Appendix I.
>
> We also consider a wider range of text anonymization tools, including other NER based ones (Scrubadub, GliNER), and *we have now also included other LLM-based approaches* (Clio, Rescriber, GPT-4.1 with the Anthropic prompt, the iterative anonymizer by Staab et al. (2025)), and *approaches based on differential privacy* (see Section 5).
>
> Utility and computational cost are also important considerations when evaluating these tools. While we do not claim novelty in how these metrics are computed, we include them as essential complements to re-identification risk: both are necessary for measuring progress and for enabling practitioners to make informed choices suited to their needs.

---

> ### Author Response · Authors · 2025-11-21
>
> > The LLM-based anonymization method used in the evaluation is, in the face of existing literature to this end, ad-hoc. The authors should have evaluated the LLM-based anonymization methods that they cite themselves as well in Section 2.
>
> Thank you for raising this point. In our benchmark, we evaluated the performance of LLMs instantiated on the PII purifier prompt, which is provided by one of the industry leaders in the field of LLM development. However, we agree that additional anonymization methods will improve the scope of our benchmark and provide useful insight into each anonymizer. To this end, we have now further expanded our evaluation to also include other LLM-based anonymization methods. We *included GPT-4.1 as a more capable anonymizer* (instantiated with the same Anthropic prompt used for Llama-3.1-8B and Gemini-2.5-Flash in Table 1), and we have *added additional prompts*, Clio and Rescriber, for Llama-3.1-8B and GPT-4.1. With the Anthropic prompt, using a more capable model such as GPT-4.1 does lead to a modest 5% reduction in re-identification risk. Using the Rescriber prompt, which lists more specific PII types, improves results further for both models, suggesting that prompt quality matters alongside model strength. Finally, Clio is the most successful at reducing re-identification risk, but at a substantial utility cost (Figure 4a) , as it often rewrites or summarizes the text rather than making minimal edits. We report all results in Section 5.
>
> We have also incorporated the *iterative anonymizer proposed by Staab et al. (2025)*, using GPT-4.1 as both the attacker and anonymizer component.
>
> We note, however, that iterative anonymizers of this kind may risk overfitting to the benchmark. Because the anonymizer has explicit access to the attacker’s feedback and can repeatedly revise the text with perfect knowledge of which attributes are being targeted, it would likely perform better in this controlled setting. The reported benchmark performance would not then measure the true real-world risk as, in practical deployments, such comprehensive knowledge of all possible identifiers, especially indirect identifiers whose importance might depend strongly on the context, is unlikely to be available.
>
> To study this, we evaluate the iterative anonymizers for the following 4 cases:
>
> - (“Ideal”) In this case, we assume the iterative anonymizer to have exact knowledge of all attributes that appear, all potential “leaks”. We therefore supply it with the full list of 9 indirect and 6 direct identifiers used to construct profiles in our benchmark. This corresponds to an ideal, fully specified scenario in which we expect the anonymizer to perform well.
>
> - (“Ideal-extended”) We then evaluate the performance when the list of target attributes given to the iterative anonymizer is made more exhaustive, reflecting a more conservative setting. We pass the anonymizer the full “Ideal” list, extended with 10 additional identifiers taken from PUMS columns that are not used in our benchmark generation.
>
> - (“Generalization”) In this setting, the anonymizer has only partial knowledge of the attributes that may appear, allowing us to test how well its performance generalizes to identifiers that are not explicitly specified. For each text instance, we randomly sample a subset of 5 identifiers from the “Ideal” list and provide only this subset to the anonymizer.
>
> - (“Out-of-the-box (Presidio)”) Finally, we consider a scenario in which the iterative anonymizer relies on a generic, pre-defined set of identifying attributes rather than one tailored to our benchmark. For this, we provide the general and US-specific identifier lists used by Microsoft Presidio.
>
> Our results (Table 2) show that the iterative anonymizer performs very well when given the full, exact set of attributes (“Ideal). Under this setting, the re-identification risk drops to 13%, lower than most anonymization tools in Table 1, while the “Ideal-extended” further reduces the risk to 10%. When the iterative anonymizer has less precise information about the target attributes, however, its effectiveness diminishes: re-identification risk is reduced only to 24% under “Generalization”, and to 45% in the “Out-of-the-box (Presidio)” configuration.
>
> These findings demonstrate that iterative anonymization as proposed by Staab et al. (2025) is a promising direction, while also highlighting the importance and difficulty of specifying its attributes for new settings. This only reinforces the importance of benchmarks like the one we propose. Other future improvements may also include better prompt design for one-shot LLM anonymizers or even fine-tuning models specifically for this task.
>
> We provide the full results in Section 5 of the revised paper.

---

> ### Author Response · Authors · 2025-11-21
>
> > The LLM adversary is a weak local model; understimating the privacy risks. Other works on LLM-based inference risks argue for strong adversaries for this reason (Staab et al. 2025).
> Thank you for raising this point. We chose Llama-3.1-8B as our attacker due to its strong performance, while being a relatively lightweight model (compared to, e.g., GPT-4), and one we are able to run locally. We agree, however, that a stronger adversary model may potentially increase the re-identification risk of anonymized texts.
> In response, we re-ran our analysis with a stronger attacker, GPT-4.1, and compared the resulting re-identification risk to that obtained with Llama-3-8B from the original submission. For computational efficiency, we focused on three anonymizers (Azure, GliNER, and Llama with the Anthropic prompt). As shown in Appendix L.1, the risks are very similar, indicating that for the difficulty levels in our benchmark, Llama-3-8B already acts as a sufficiently strong attacker.
> > The introduced benchmark dataset is not as realistic, robust, and versatile as datasets used in other works. Unclear why these were not employed, or why their synthetic construction quality criteria were not observed (e.g., Yukhymenko et al. 2024).
>
> A key component of our benchmark is to accurately measure re-identification risk in anonymized texts. We achieve this by grounding our synthetic data generationin real-world US demographics, such that any re-identification risk resulting from correctly inferred indirect identifiers can be computed using the method of Rocher et al. (2019).  As SynthPAI from Yukhymenko et al. (2024) uses fully synthetic profiles, it could not have been adapted for this purpose.
>
> We further analyze the realisticness and diversity of our dataset by:
> - Conducting a human annotation experiment for each level of difficulty on 180 randomly selected benchmark samples. We find agreement between a blind human annotator and the synthetic generator in 94% of studied cases. Further results are provided in Appendix E.2.
> - We study the diversity of attributes we include in the text in Appendix E.1.
> - We provide examples of benchmark entries for both scenarios, and all levels of difficulty in Appendix D.
>
> > The figures on page 8 are rather small and difficult to read.
>
> Our apologies. We have now increased the size of Figure 4. For the utility analysis, we also provide additional results in Appendix M.
>
> > The x-axis in Figure 1 (a) is unclear. How can one have a monotonous function (ordered) over these categories (independent)?
>
> Our apologies if Figure 1(a) was unclear. Its purpose is to further illustrate our computation of re-identification risk from Algorithm 2. In this example, an attacker attempts to re-identify an individual from 2 anonymized texts by sequentially trying to recover any personal attributes, starting with the 5 direct identifiers, followed by the 9 indirect identifiers we consider. We plot the re-identification risk as the attacker aims to sequentially infer the attributes.
>
> In the first case (orange), the attacker is able to correctly identify one direct attribute, which immediately sets the risk to 1; subsequent attributes are therefore irrelevant. In the second case (blue), no direct identifier is recovered, so the attacker proceeds to infer indirect attributes. The risk then increases in a stepwise fashion each time an additional indirect attribute is correctly inferred. Note that the values in this case are for illustrative purposes only. We have further clarified the figure caption.
>
> **We hope these revisions address your concerns. Please let us know if there are any remaining issues, suggestions, or questions that -- if clarified or addressed -- would contribute to raising the overall score.**

---

> ### Author Response · Authors · 2025-11-21
> **References**
>
> [1] R Staab et al. Beyond memorization: Violating privacy via inference with large language models. ICLR 2024.
>
> [2] H Yukhymenko et al. A Synthetic Dataset for Personal Attribute Inference. NeurIPS 2024.
>
> [3] R Staab et al. Large language models are advanced anonymizers. ICLR 2025.
>
> [4] Brown, Hannah, et al. "What does it mean for a language model to preserve privacy?." ACM conference on fairness, accountability, and transparency. 2022.
>
> [5] Liu, Yupei, et al. "Evaluating {LLM-based} Personal Information Extraction and Countermeasures." (USENIX Security 25)
>
> [6] Patsakis, Constantinos, and Nikolaos Lykousas. "Man vs the machine in the struggle for effective text anonymisation in the age of large language models." Scientific Reports.
>
> [7] Pilán, Ildikó, et al. "The text anonymization benchmark (tab): A dedicated corpus and evaluation framework for text anonymization." Computational Linguistics 48.4 (2022)
>
> [8] Rocher, Luc, Julien M. Hendrickx, and Yves-Alexandre De Montjoye. "Estimating the success of re-identifications in incomplete datasets using generative models." Nature communications.
>
> [9] https://openai.com/en-GB/policies/how-your-data-is-used-to-improve-model-performance/
>
> [10] https://techcrunch.com/2025/08/28/anthropic-users-face-a-new-choice-opt-out-or-share-your-data-for-ai-training/
>
> [11] Jennifer King, Kevin Klyman, Emily Capstick, Tiffany Saade, and Victoria Hsieh. User privacy
> and large language models: An analysis of frontier developers’ privacy policies.
>
> [12] https://www.reuters.com/business/media-telecom/openai-fights-order-turn-over-millions-chatgpt-conversations-2025-11-12/
>
> [13] Devansh Singh and Sundaraparipurnan Narayanan. Unmasking the reality of pii masking models: Performance gaps and the call for accountability.
>
> [14] Zhou, W., Zhang, S., Gu, Y., Chen, M., & Poon, H. (2023). UniversalNER: Targeted Distillation from Large Language Models for Open Named Entity Recognition. In ICLR 2024

---

> ### Comment · Reviewer_8xbg · 2025-11-24
>
> Thank you for the detailed rebuttal, however, my main concerns remain.
>
> The benchmark does not live up to the current standards in the literature.
>
> The fact that stronger attacker models do not achieve stronger results than Llama 3 8B is strange; especially given the fact that we know from other works that frontier models outperform Llama 3 8B on attribute inference.
>
> The fact that the benchmark is grounded in US demographics is neat, but: (i) for the individual reidentification risk, this does not matter, this is always accurately calculatable for a set of attributes given the necessary demographic information; and (ii) the SynthPAI framework could still be used to generate synthetic data grounded in any demographics by adjusting the initial distribution of the profiles.
>
> Finally, while it is true that from the already pooled accuracy scores of eg Staab et al. (2024) one cannot directly calculate the reidentification risk, given the individual results (ie the set of accurately inferred attributes per profile) this can be done by applying the framework of Rocher et al. (2019). The fact that the authors actually do this, and argue for this metric, is again, neat and valuable, but alone does not mean a publication-worthy contribution in the current framework of this paper, especially given the limitations of the dataset. Also, as the difficulty of the synthetic dataset is entirely ungrounded, the actual final reidentification results are just as much synthetic as they would be if the initial attribute distribution was not grounded in real demographic data. Merely, the model inferences are a <1.0 multiplier on the intrinsic maximal reidentification risk that stems from the pre-hoc construction of each sample and the planted identifiers. As such, there is really no difference in value in calculating this metric for SynthPAI or for the proposed dataset. However, if the dataset would consist of actual real-world data, the insights would be practical.

---

> > ### Author Response · Authors · 2025-11-28
> > **Answer to the reviewer**
> >
> > We thank the reviewer for engaging with our rebuttal. We hope that our new additions,  including the analysis of the iterative anonymizer (Section 5), the use of more state-of-the-art models for generation (Appendix B) and for attacking (Appendix L.1), and a more in-depth examination of anonymizer failure cases (Appendix I) were helpful. Below, we provide **further clarification on the points raised** and conclude with a **summary of our contributions**.
> >
> > >  The fact that the benchmark is grounded in US demographicsas is neat, but: (i) for the individual reidentification risk, this does not matter, this is always accurately calculatable for a set of attributes given the necessary demographic information; and (ii) the SynthPAI framework could still be used to generated synthetic data grounded in any demographics by adjusting the initial distribution of the profiles.
> >
> > Our apologies if this was not clear. To be able to compute a *real-world re-identification risk* for a given population from a set of indirect identifiers, one needs:
> > - *Attribute encodings that match the real world.* The way attributes are defined and encoded must align with how they appear in reality.
> > - *Representative individual-level data (or carefully designed aggregate statistics)* The risk indeed depends on the full joint distribution of attributes across individuals, not just their marginals.
> > - *Profiles that are representative of the target population*. This again requires preserving realistic combinations of attributes, rather than sampling from marginals.
> >
> > Our setup incorporates all of these necessary elements **by design**, and they cannot easily be added post-hoc. In particular, purely **synthetic indirect attributes**, such as those used in SynthPAI, **typically do not match real-world attribute encodings, nor do they capture realistic joint distributions**. Moreover, the correct inference of an attribute is not equally important for each attribute, or individual. For instance, some real-world attribute combinations inherently carry higher re-identification risk (e.g., a 30-year-old personal trainer is far less unique than a 60-year-old one), which relies on faithful modelling of real attribute combinations.
> >
> > **We agree that the SynthPAI framework could, in principle, be extended toward our use case, but only if it were (i) seeded with the same real-world indirect identifiers as we use, (ii) augmented with direct identifiers in the way we propose, and (iii) used to compute re-identification risk according to our methodology. This could, eventually, yield a Reddit-style variant of our scenarios. However, we view these changes as *substantial* and hence consider our work as *complementary* to the otherwise very meaningful work of SynthPAI, rather than proposing a set of features that can be easily added on top of it.**
> >
> > > actualy final reidentification results are just as much synthetic as they would be if the initial attribute distribution was not grounded in real demographic data. Merely, the model inferences are a <1.0 multiplier on the intrinsic maximal reidentification risk that stems from the pre-hoc construction of each sample and the planted identifiers.
> >
> > We agree with the reviewer that maximal re-identification risk depends, by definition, on both the number,  and the type and values of identifiers present in the anonymized text. While we do not argue that an average real medical consultation or chatbot conversation contains exactly one direct and five indirect identifiers (as in Table 2), our setup is designed to correctly measure the re-identification risk if that were the case. We then **systematically vary the number and type** of identifiers and show how the risk evolves in Figures 2 and 3.
> >
> > We view this controlled leakage of a certain number of identifiers as a **key contribution, as attribute inference accuracy cannot be meaningfully aggregated across multiple attributes into a single re-identification risk value**. For example, our setup naturally handles attributes with highly skewed distributions or only a few possible categories: such attributes may be easy for an attacker to predict (and thus lead to high *inference accuracy*) but contribute only marginally to the overall *re-identification risk*. Likewise, if feature values are fully synthetic and do not capture realistic *joint* feature distributions, any resulting estimate of re-identification risk, even if computed as in our setup, would not be meaningful.

---

> > > ### Author Response · Authors · 2025-11-28
> > > **Summary of contributions**
> > >
> > > While we agree with the reviewer that Yukhymenko et al. (2024), similar to previous open source datasets such as Pilán et al. (2022) and Singh & Narayanan (2025), is a valuable contribution for Personal Attribute Inference research, we believe that for the purpose of benchmarking text anonymizers, our work offers **several key contributions on top of the existing literature** (including SynthPAI). We summarize them below:
> > >
> > > | Characteristic | SynthPAI | RAT-Bench | Importance |
> > > |----|----|----|---|
> > > | Goal | Generate synthetic data to **evaluate attribute inference by LLMs** (18 LLMs evaluated) | Propose an **end-to-end benchmark to evaluate anonymization tools**, measuring progress and identifying failure modes (13 tools evaluated) | Both goals are important, yet meaningfully different. |
> > > | Evaluation | Based on **inference accuracy** of features in anonymized text | Introduces a method to compute **re-identification risk** of anonymized text, based on values of inferred direct and indirect identifiers |  To evaluate anonymization, the risk of re-identification needs to be considered from both direct and indirect identifiers, in line with legal standards |
> > > | Tested Identifiers | **8 indirect** identifiers | **6 direct, 9 indirect** identifiers | Rigorous benchmarking requires anonymizers to be tested on an **extensive list of both direct and indirect identifiers** |
> > > | Source for attributes | Fully **synthetic attributes**  | Indirect identifiers grounded in **real-world demographics** with synthetic direct identifiers |  Realistic joint distributions are necessary to compute re-identification risk|
> > > | Feature insertion method| **Post-hoc analysis** of generated synthetic data, resulting in uneven distribution of features and difficulty level across dataset | **Directly and systematically** inserts **diverse** set of feature combinations across **all difficulty levels** |  Diverse range of feature combinations across difficulty levels **enables rigorous benchmarking** of anonymizers and measurement of progress |
> > > | Scenarios | 1 scenario (**Reddit** comment thread) | 2 scenarios (**Medical consultation** transcript, **chatbot interaction**) | Evaluate performance of anonymizers across multiple settings, in line with recent developments |
> > >
> > > We thank the reviewer for their time and quick response, and hope they find our work meaningfully contributing to the field. We would be glad to incorporate any further suggestions that could help address the reviewer’s remaining concerns.

---

### Official Review · Reviewer_bkRV · 2025-10-30

**Soundness:** 2
**Presentation:** 3
**Contribution:** 3
**Rating:** 6
**Confidence:** 4

**Summary:**

This paper introduces the first modern synthetic benchmark for evaluating text anonymization tools by measuring their effectiveness at preventing re-identification of individuals in the U.S. population. The authors generate realistic text grounded in demographic data containing both direct identifiers (names, SSNs) and indirect identifiers (occupation, state), apply seven anonymization tools (NER-based and LLM-based), and use an adversarial LLM to infer attributes from the anonymized text.

**Strengths:**

1) The paper introduces the first benchmark that measures anonymization effectiveness through actual re-identification risk rather than just entity recall, combining both direct and indirect identifiers with a rigorous methodology grounded in real U.S. demographic data (PUMS) and the established re-identification framework from Rocher et al. (2019).
2) The benchmark design is thorough, incorporating controlled difficulty levels (easy/medium/hard), diverse scenarios (medical transcripts, chatbot interactions)

**Weaknesses:**

* The paper misses several important strong text-to-text privatization baselines [1,2,3]. Listing them here, would highly recommend discussing and if possible adding them as baselines to properly support the claim "The best anonymizer leaves a significant re-identification risk of 36% in our setup"


* Did the authors try to consider multiple threat models, such as the Static Attacker and Adaptive Attacker in [4]?


Refs

[1] Privacy-and utility-preserving textual analysis via calibrated multivariate perturbations .WSDM 2020

[2] TEM: High Utility Metric Differential Privacy on Text, SIAM, 2023

[3] Locally differentially private document generation using zero shot prompting, EMNLP 2023

[4] The Limits of Word Level Differential Privacy, EMNLP 2022.

**Questions:**

NA

---

> ### Author Response · Authors · 2025-11-21
> **Response to Reviewer bkRV**
>
> We thank the reviewer for their time spent on the paper and their thoughtful feedback. Below, we address each of the points raised.
>
> > The paper misses several important strong text-to-text privatization baselines [1,2,3].
>
> We thank the reviewer for this suggestion. Our benchmark currently focuses on methods that remove or mask the specific text spans containing identifying information, whereas the reviewer highlights perturbation-based approaches that modify the entire text by adding controlled noise. While these methods may not be directly comparable, we agree that they represent an interesting avenue of research in text anonymization.
>
> Hence, we now also evaluate Madlib [1], TEM [2] and DP-Prompt [3] in our benchmark. We discuss the results and implementation details in Appendix N, and incorporate the best performing method (TEM for epsilon=11) in Figure 1b, Table 1 and Figure 4. We also provide the corresponding BLEU scores. As expected, we find that these are substantially lower than other methods, as these methods perturb the entire sequence rather than selectively targeting spans containing identifying information. As a result, BLEU scores may not be directly  comparable to those of the other methods in our benchmark. We leave the development of a utility metric that more fairly compares these approaches to future work. We now also discuss these methods in Section 2.

---

> ### Author Response · Authors · 2025-11-21
>
> > Did the authors try to consider multiple threat models, such as the Static Attacker and Adaptive Attacker in [4]?
>
> We thank the reviewer for this suggestion. We agree that considering a threat model in which an attacker has full knowledge of the anonymization method (Adaptive in [4]) versus not (Static in [4]) is an important and valuable distinction, especially as the anonymization tools evaluated in this work are broadly available. However, we argue that this would give the attacker less advantage in our setup than the one considered in [4] (see below), and also confirm this in a proof-of-concept experiment.
>
> Mattern et al. [4] evaluates how word-level perturbation techniques protect privacy through the performance of a BERT-based *authorship classifier* for 10 authors. For the Static attacker, the classifier is trained on the original text, while for the Adaptive attacker, it is trained on perturbed text. Both are then evaluated on a test set of perturbed text. They show that, while perturbations easily fool the static classifier, significant author-specific information can still be picked up by an adaptive classifier.
>
> In our setting, we do not attempt to classify authorship and, instead, evaluate anonymization tools by asking an LLM-based attacker to *infer attribute values* (e.g., phone number, state of residence) from anonymized text, and then translate the correctly inferred attributes into a re-identification risk.
>
> This means that, in our setup, we indeed assume a *static* attacker, as they do not know or leverage the anonymization mechanism. We agree with the reviewer that the threat model of an adaptive attacker is relevant, as anonymization mechanisms should not be considered a secret in practice. However, because our attacker’s task is to recover concrete attributes (which are either fully removed or still inferable from the anonymized text), we believe that knowing the exact anonymization method (e.g. whether it was NER-based or using Gemini) would not substantially simplify the task.
>
> To confirm this, we instantiate a proof-of-concept *Adaptive* attacker for 2 anonymization methods. For each, we select 3 original benchmark entries, and their anonymized version, and provide these as additional context to the attacker LLM. As such, the attacker is aware of the anonymization pattern through in-context learning. We evaluate this on the remaining benchmark entries for the medical conversation scenario for each level of difficulty and report the re-identification success rate (%) for both the static and the adaptive attacker:
>
> | Anonymization tool  | Difficulty Level | Static Attacker | Adaptive Attacker |
> |----|----|----|----|
> | Azure  | Level 1  | 28 | 26 |
> |   | Level 2  | 32 | 32  |
> |   | Level 3 | 27| 17 |
> | Anthropic prompt + Llama | Level 1 | 47 | 42|
> |  | Level 2 | 41 | 43 |
> |  | Level 3 | 32 | 22 |
>
> We find that the re-identification rate for the adaptive attacker (given example anonymized text in-context) either remains similar to, or slightly lower than, that of the static attacker. This suggests that knowing the anonymizing pattern either offers limited benefit, or that the provided examples may even confuse the LLM-based attacker. We leave for future work to explore how an LLM-based attacker, prompted to infer attributes, could further leverage knowledge of the exact anonymization method to improve its inference success.
>
> Overall, we thank the reviewer for the thoughtful suggestion and have added these results and discussion to Appendix L.2.
>
> **We hope these revisions address your concerns. Please let us know if there are any remaining issues, suggestions, or questions that -- if clarified or addressed -- would contribute to raising the overall score.**

---

> > ### Comment · Reviewer_bkRV · 2025-11-22
> > **Thanks for the response**
> >
> > I thank authors for the detailed response and additional experiments! I have now increased my score.

---

### Official Review · Reviewer_uffD · 2025-10-30

**Soundness:** 3
**Presentation:** 3
**Contribution:** 2
**Rating:** 4
**Confidence:** 5

**Summary:**

This paper proposes a new dataset and benchmark for evaluating how well current PII anonymizers and text sanitizers perform at actually removing direct and indirect attributes from written texts. In particular, the presented dataset of 600 samples (across three hardness levels) consists of two main settings: (1) fictional doctor-patient conversations and (2) fictional AI chatbot interactions. The authors measure how well an adversary (instantiated via an LLM) can recover these direct and indirect identifiers both before and after anonymization. For indirect identifiers, the likelihood of unique identifiability (based on the US Census) is reported. On their dataset, they benchmark five NER-based anonymizers and one version of LLM-based anonymization (in two model variants), finding that overall Azure performs best across both settings and hardness levels. Typical trends across hardness levels are also ablated, and overall, in any level or setting, the final re-identification success rate remains very high ($\geq 24\%$). While generally faster and more thorough than the tested LLM versions, NER anonymizers have a higher impact on the utility of the resulting texts (as reported via BLEU and ROUGE).

**Strengths:**

- The overall issue of existing anonymizers falling short of anonymizing texts is an actual problem with real-world implications for people's privacy. Any effort in this direction is worth it in the reviewer's opinion. Existing industry-level anonymizers should not miss "an average of 4.8%" of direct identifiers.
- The general split of hardness levels is nice and reveals intuitive patterns. Overall, having datasets to evaluate the performance of methods on is important for the field. Further, the two scenarios are generally not unrealistic.
- The inclusion of the re-identification risk via indirect attributes seems a good way of contextualizing non-direct privacy leakage in this setting.
- The failure cases found this way (App. E) can (hopefully) help to improve the underlying anonymizers themselves.

**Weaknesses:**

- The individual samples are, in parts, quite unrealistic (in particular, level 2). This by itself is not an issue (for PII removal, it should work statically on any data source); however, it is problematic when we try to draw conclusions about the real-world impact of the lack of proper anonymization. With this, the dataset has more relevance as an individual "how well are removal engines in these synthetic scenarios" benchmark.
- This goes hand in hand with these samples being generated by a quite weak model (2.5-flash) and not having undergone any human validation (at least not reported). While the samples I saw were generally fine (although quite artificial), prior work in this area either used real-world data [1,3] or released human-verified samples alongside [4,5].
- The ground-truth adversary was chosen as an 8B LLM. This is particularly problematic, as prior work has shown that attribute or identifier inference correlates strongly with model capabilities. As such, the currently reported numbers serve merely as a rough lower bound, which likely could be considerably higher. This is in part even acknowledged by "but rather because the attacker LLM itself struggles more to correctly infer attributes even without anonymization," which generally holds but much more so with weaker adversaries.
- The instantiated LLM anonymizers based on the Anthropic PII perform worse than Azure. At the same time, prior work (e.g., [4]) has already shown that stronger anonymization via LLMs is possible, raising the question of why such stronger methods were not evaluated (in particular, many of the missed instances in Table 6 seem like they should be detectable there). Similarly, the current LLM anonymizers were instantiated with comparatively weak models, which can make a big difference in anonymization.
- The "being first to do it" angle seems a bit oversold to the reviewer given that prior work already has generated at least equally realistic synthetic benchmarks for anonymization [5] that could be directly adapted to the re-identification setting by applying the corresponding definition from

**Questions:**

Besides the above, I have the following questions:

- Not many dataset statistics are given in the work. What is the distribution of these identifiers for the numbers you report? How are they distributed over hardness levels? How are they co-distributed in texts and potentially correlated? So far, I can only roughly estimate some marginals from the recall numbers in Figure 5 and Figure 6.
- Do you have general insights on which types of failures (FP and FN) we observe per model type?
- How do you interpret your current results with respect to prior work that shows LLMs clearly outperforming methods like Azure or Presidio [2,4,5]? Which conclusions should practitioners, users, and anonymization system developers draw when they see both that Azure is insufficient across much prior work and the statement that they "provide computationally efficient protection" (on synthetic data)?
- Smaller: Matching based on Jaro-Winkler for free-text attributes is an improvement over direct matching. Did you manually verify that this threshold holds here? From personal human post-labeling efforts, this can still be partly off.

Sources

[1] Pilán, Ildikó, et al. "The text anonymization benchmark (tab): A dedicated corpus and evaluation framework for text anonymization." _Computational Linguistics_ 48.4 (2022): 1053-1101.\
[2] Bubeck, Sébastien, et al. "Sparks of artificial general intelligence: Early experiments with gpt-4." _arXiv preprint arXiv:2303.12712_ (2023).\
[3] Staab, Robin, et al. "Beyond memorization: Violating privacy via inference with large language models." _arXiv preprint arXiv:2310.07298_ (2023).\
[4] Staab, Robin, et al. "Large language models are advanced anonymizers." _arXiv preprint arXiv:2402.13846_ (2024).\
[5] Yukhymenko, Hanna, et al. "A synthetic dataset for personal attribute inference." _Advances in Neural Information Processing Systems_ 37 (2024): 120735-120779.\
[6] Luc Rocher, Julien M Hendrickx, and Yves-Alexandre De Montjoye. Estimating the success of
re-identifications in incomplete datasets using generative models. Nature communications, 10(1):
3069, 2019.

---

> ### Author Response · Authors · 2025-11-21
> **Response to Reviewer uffD**
>
> We thank the reviewer for their time spent on the paper and their thoughtful feedback. Below, we address each of the points raised.
>
> > The individual samples are, in parts, quite unrealistic (in particular, level 2). This by itself is not an issue (for PII removal, it should work statically on any data source); however, it is problematic when we try to draw conclusions about the real-world impact of the lack of proper anonymization.
>
> We thank the reviewer for raising this very important point.
>
> The goal of our benchmark is to provide a standardized, objective, and reproducible way to measure and compare the performance of different anonymizers on a range of tasks, from easy to hard ones. We hope for it to allow developers to meaningfully measure progress, improve their methods, and to allow users to select the best anonymizer for their task.
>
> Importantly, we agree with the reviewer that the risks of re-identification we report are not meant to measure the *average* risk for a medical record. For instance, by averaging the risk across the levels 1, 2, and 3, we do not suggest that these occur with equal frequency in real-world data; we still expect e.g. Level 2 (non-standard mentions) to be substantially less common than the explicit occurrences (Level 1). However, even if Level 2 samples are rare in real-world settings, we would still expect that if an attacker can infer the attributes, an anonymization tool should be capable of removing it. Similar to other benchmarks designed to evaluate e.g. reasoning abilities of frontier LLMs [7]: not all of their more complex problems are equally likely to occur in practice, yet their inclusion in benchmarks remains important for measuring progress.
>
> We also now verify the presence of Level 2 examples in real-world datasets. To do so, we randomly selected 10,000 emails containing more than 300 words from the Enron email corpus (https://huggingface.co/datasets/corbt/enron-emails), and used LLaMA-3-8B-Instruct to search for occurrences of attributes corresponding to Levels 1, 2, or 3. As expected, we observe many Level 1 occurrences and considerably fewer Level 2 and Level 3 occurrences. We include a selection of representative examples below.
>
> Level 2:
> - Address: "four hundred and fifty five on Oak Hill Road"
> - State of residence: "NYC"
> - Age: "Harris is about 14mos now"
> - Gender: "Religious guy"
> - Phone number: "3-3859" (phone extension within a company)
> - Phone number: "(713) * 993-5515"
> - Phone number:: "713.584.4444"
>
> Level 3:
> - Marital status: "I've gone out on a couple of dates"
> - US citizenship status: "Australian"
> - Age: "freshman"
> - Education level: "Senior Lecturer"
> - Address: "ETOL administration building on the Wilton International site"
>
> While these examples are, of course, anecdotal, they were collected quickly from a limited set of real-world data, demonstrating that such cases do arise in practice.

---

> ### Author Response · Authors · 2025-11-21
>
> > This goes hand in hand with these samples being generated by a quite weak model (2.5-flash) and not having undergone any human validation (at least not reported).
>
> We agree that Gemini-2.5-Flash may not be state-of-the-art, but we selected it as a cost-efficient model that would still provide sufficient capability to generate diverse benchmark entries. We now verify whether more capable models generate better benchmark entries.
>
> To do so, we use GPT-5 to generate 150 additional benchmark entries and compare the performance of a subset of anonymizers (Azure, GliNER, and Llama with the Anthropic prompt) on the GPT-5-generated benchmark data to the original Gemini-generated benchmark.
>
> Interestingly, Figure 5 in Appendix B shows that there is no significant difference between the GPT-5 and Gemini benchmarks in re-identification risk of anonymized texts across all three anonymizers. This suggests that producing synthetic data with controlled mention of identifying attributes is a task for which Gemini is already sufficiently capable, and that the additional capabilities of a stronger model like GPT-5 does not lead to meaningfully different results for the purposes of evaluating re-identification risk. From manual inspection, we do find that synthetic records generated by GPT-5 are slightly more realistic and of higher quality than those generated by Gemini. We provide some examples of the generated texts in Appendix D.2.
>
> > The ground-truth adversary was chosen as an 8B LLM.
>
> We chose Llama-3.1-8B as our attacker due to its strong performance, while being a relatively lightweight model (compared to, e.g., GPT-5), and one we are able to run locally. We agree, however, that a stronger adversary model may potentially increase the re-identification risk of anonymized texts.
>
> We now include additional experiments with GPT-4.1 as the adversary on texts anonymized by Azure, GliNER, and Llama with the Anthropic prompt. We select GPT-4.1 as we found GPT-5 (at the time of running this experiment, the most recent and strongest GPT model) often refuses to infer attributes, regardless of whether they are generally considered sensitive (e.g., SSN, credit card number) or not (e.g., occupation).
>
> We find that the stronger model does not meaningfully impact the resulting re-identification risk, indicating that, for the purposes of this benchmark, Llama-3.1-8B is a sufficient attacker. The full results are included in Appendix L.1 of the updated paper.

---

> ### Author Response · Authors · 2025-11-21
>
> > The instantiated LLM anonymizers based on the Anthropic PII perform worse than Azure. At the same time, prior work (e.g., [4]) has already shown that stronger anonymization via LLMs is possible
>
> We thank the reviewer for this point. We now include both one-shot LLM anonymizers with a stronger model (GPT-4.1) and additional prompts (Clio, Rescriber), and four variations of the iterative attacker introduced by Staab et al. [4] with the same model.
>
> *One-shot LLM anonymizer*
>
> We include the new results for one-shot anonymizers in Table 1 in our updated paper. With the Anthropic prompt, using a more capable model such as GPT-4.1 does lead to a modest reduction in re-identification risk in most cases. Using the Rescriber prompt, which lists more specific PII types, improves results further for both models, suggesting that prompt quality matters alongside model strength. Finally, Clio is the most successful at reducing re-identification risk, but at a substantial utility cost, as it summarizes texts into a sentence or short paragraph rather than minimally editing it.
>
> *Iterative anonymizer*
>
> We have also incorporated the iterative anonymizer proposed by [4]. Specifically, we use GPT-4.1 as both the attacker and anonymizer component, following the implementation described by [4].
>
> We note, however, that iterative anonymizers of this kind may risk overfitting to the benchmark. Because the anonymizer has explicit access to the attacker’s feedback and can repeatedly revise the text with perfect knowledge of which attributes are being targeted, it would likely perform better in this controlled setting. The reported benchmark performance would not then measure the true real-world risk as, in practical deployments, such comprehensive knowledge of all possible identifiers, especially indirect identifiers whose importance might depend strongly on the context, is unlikely to be available.
>
> To study this, we further evaluate the iterative anonymizer for the following 4 cases:
>
> - (“Ideal”) In this case, we assume the iterative anonymizer to have exact knowledge of all attributes that appear, all potential “leaks”. We therefore supply it with the full list of 9 indirect and 6 direct identifiers used to construct profiles in our benchmark. This corresponds to an ideal, fully specified scenario in which we expect the anonymizer to perform well.
>
> - (“Ideal-extended”) We then evaluate the performance when the list of target attributes given to the iterative anonymizer is made more exhaustive, reflecting a more conservative setting. We pass the anonymizer the full “Ideal” list, extended with 10 additional identifiers taken from PUMS columns that are not used in our benchmark generation.
>
> - (“Generalization”) In this setting, the anonymizer has only partial knowledge of the attributes that may appear, allowing us to test how well its performance generalizes to identifiers that are not explicitly specified. For each text instance, we randomly sample a subset of 5 identifiers from the “Ideal” list and provide only this subset to the anonymizer.
>
> - (“Out-of-the-box (Presidio)”) Finally, we consider a scenario in which the iterative anonymizer relies on a generic, pre-defined set of identifying attributes rather than one tailored to our benchmark. For this, we provide the general and US-specific identifier lists used by Microsoft Presidio.
>
> Our results (Table 2) show that the iterative anonymizer performs very well when given the full, exact set of attributes (“Ideal). Under this setting, the re-identification risk drops to 13%, lower than most anonymization tools in Table 1, while the “Ideal-extended” further reduces the risk to 10%. When the iterative anonymizer has less precise information about the target attributes, however, its effectiveness diminishes: re-identification risk is reduced only to 24% under “Generalization”, and to 45% in the “Out-of-the-box (Presidio)” configuration.
>
> These findings demonstrate that iterative anonymization is a promising direction, while also highlighting the difficulty of specifying its attributes for new settings and importance of proper evaluation. We leave for future work to explore how the generalizability of the iterative anonymizer can be further improved. Other future improvements may also include better prompt design for one-shot LLM anonymizers or even fine-tuning models specifically for this task.
>
> We provide the full results in Section 5 of the revised paper.

---

> ### Author Response · Authors · 2025-11-21
>
> > The "being first to do it" angle seems a bit oversold to the reviewer given that prior work already has generated at least equally realistic synthetic benchmarks for anonymization [5] that could be directly adapted to the re-identification setting by applying the corresponding definition from
>
> We thank the reviewer for raising this point. There is indeed extensive and valuable work in this area, including [5], which we build on and cite throughout the paper, particularly in Section 2. We believe our contribution offers a complementary perspective to this literature (see below), and we will ensure that the manuscript’s wording reflects this more clearly.
>
> We highlight several key differences with the existing literature, including [5], that we believe are meaningful:
> - *Re-identification risk.* Prior work reports attribute-level inference accuracy. However, this does not directly reflect re-identification risk, as e.g. different attributes vary in difficulty and importance, and multiple attributes can co-occur in a single text, potentially amplifying risk in ways that accuracy cannot capture. Instead, we map correctly inferred identifiers to re-identification risk. Our approach incorporates both direct and indirect identifiers, each of which contributes differently and must be considered from a legal perspective, and systematically varies their co-occurrence within a single text.
> - *Grounded in real-world demographics.* When only indirect identifiers are correctly inferred, we use Rocher et al.’s [6] estimator to compute the re-identification risk for the US population. Incorporating real-world sets of indirect identifiers in constructing the benchmark, rather than synthetic ones [5], is essential for producing realistic, population-level estimates of re-identification risk.
> - *Systematic control.* We systematically vary the number and type of direct or indirect identifiers leaked according to varying levels of difficulty. This enables rigorous benchmarking of a wide range of anonymization techniques, measuring their progress and evaluating their failure modes.
> - We consider two distinct *scenarios*: medical consultations and AI-chatbot interactions. The latter is increasingly relevant given that AI companies use user-chat data for model improvement [8,9,10] and, just last week, a court required OpenAI to release anonymized chat transcripts [11].
>
> > Not many dataset statistics are given in the work. What is the distribution of these identifiers for the numbers you report? How are they distributed over hardness levels? How are they co-distributed in texts and potentially correlated? So far, I can only roughly estimate some marginals from the recall numbers in Figure 5 and Figure 6.
>
> Our apologies if this was not clear. For our main benchmark entries (used in Table 1), we generate 100 records for each scenario and level of difficulty. For each of these entries, we randomly sample 1 out of the 5 target direct identifiers to be included, and 5 out of the 9 indirect identifiers to be included. We have now clarified this in Section 4 and we now visualize the number of occurrences per attribute in each level in Appendix E.1.

---

> ### Author Response · Authors · 2025-11-21
>
> > Do you have general insights on which types of failures (FP and FN) we observe per model type?
>
> This is a very interesting question! Such insights would really help guide future improvements. We now add a full list of insights (FP and FN) for each anonymizer in Appendix I, and provide a short summary here below.
>
> For NER-based approaches, we find that many false positives come from text spans that resemble named entities but are not actually identifying. For example, Azure often removes generic person nouns such as “patient,” and GliNER may remove pronouns like “I” or “me and my buddies.” Similarly, Presidio sometimes deletes common temporal expressions such as “today”, which on their own are unlikely to constitute an indirect identifier. These methods seem to be confusing these pieces of text with what is frequently annotated as PERSON or TIME/DATE entities in standard NER datasets, which are categories that can include names or dates of birth, but also many non-identifying terms. As a result, NER-based anonymizers may over-redact text that does not meaningfully contribute to re-identification risk.
>
> We also find that many false negatives in NER-based models arise when identifiers appear in non-standard forms, such as being split across multiple spans (e.g., “My phone number is 312, then 480, then 3820”) or expressed through slang (e.g., “C-way” for Conway). This suggests that many NER models are optimized for detecting entities in their standard forms and struggle with variability in this format. Finetuning these models on datasets with annotated non-standard language, e.g. transcripts, may help address this gap. Beyond that, we also find that NER-based methods can sometimes also miss clearly stated identifying information, with for instance Scrubadub missing clear instances of SSNs or credit card numbers, or Uniner not removing all duplicates of the same identifier in a piece of text. We hypothesize that such failures stem from surrounding context that differs from the ones seen during training, but we leave a deeper investigation to future work.
>
> When it comes to LLM-based anonymizers we find them to have less false positives than NER-based methods, resulting in better utility (Figure 4a). For instance, we find them to be more precise in distinguishing genuinely identifying (e.g. a person’s name) information from entity-like but harmless text (e.g. ‘patient’). We also find that LLM-based anonymizers tend to perform better with unusual representations, with significantly lower false negative rates on difficulty level 2 records than NER models. This indicates that LLMs use reasoning and a broader contextual understanding to distinguish what is truly identifying information and interpret fragmented or unconventional patterns, compared to NER models which primarily match patterns seen during training, making LLM-based anonymization a promising area of research.
>
> Finally, for false negatives, we find that LLMs such as Gemini and Llama instantiated with the Anthropic prompt also show inconsistencies, removing an identifier in one instance but missing it in an otherwise similar case. We also observe that LLM anonymization performance is sensitive to both model capabilities and prompt design. For the same prompt, a more capable model like GPT-4.1 reduces re-identification risk more effectively than Llama-3.1-8B, and both models perform better when using the more specific Rescriber prompt (Table 1). Similarly, we find that the exact attributes mentioned in the iterative anonymizer also heavily impacts performance (Table 2). These findings suggest that effective LLM-based anonymization requires carefully balancing model capability with thoughtful prompt design.

---

> ### Author Response · Authors · 2025-11-21
>
> > How do you interpret your current results with respect to prior work that shows LLMs clearly outperforming methods like Azure or Presidio [2,4,5]? Which conclusions should practitioners, users, and anonymization system developers draw when they see both that Azure is insufficient across much prior work and the statement that they "provide computationally efficient protection" (on synthetic data)?
>
> That is a great question. We do believe that choosing the right anonymizer for a given use-case requires balancing trade-offs in privacy, utility, and computational cost.
>
> Our results generally agree with prior work that LLM-based anonymizers provide a stronger privacy-utility trade-off. They remove identifying information in a more precise manner, allowing  methods such as GPT-4.1 instantiated with the Anthropic prompt to substantially reduce re-identification risk while maintaining high BLEU scores. However, these models are (computationally) expensive and might not be feasible to run at scale. We leave for future work to explore how smaller LLMs, instantiated with a carefully crafted prompt, or potentially finetuned to remove identifying information, could offer similar performance while reducing cost.
>
> In contrast, more light-weight methods, such as Azure, are computationally efficient and may reduce the re-identification more substantially, but often at a cost in utility, at least as measured by BLEU or ROUGE scores. Depending on the application, this may or may not be acceptable: in some cases, (over-)aggressive removal is harmless, while in others the semantic utility might be more important. In the latter case, approaches like Clio, which summarizes text while removing identifying information to maintain overall semantic meaning, can be more appropriate.
>
> When it comes to privacy, the choice likely also depends on the likely prevalence of identifying information and on the tolerance for false negatives. If the application requires *all* (including e.g. rare occurrences of identifiers in unusual formats) identifiers to be removed, a carefully designed iterative LLM-based anonymizer might be required, and relying solely on NER-based anonymizers may be insufficient.
>
> For anonymization system developers, our results point to several directions for future work. First, we believe more emphasis should be placed on indirect identifiers. A significant proportion of the re-identification risk in our benchmark comes from indirect identifiers, many of which are often missed by all tested anonymizers (Figure 2). Second, NER-based anonymization tools should be more robust to unusual representations of identifiers, likely requiring new annotated datasets that capture such variability. Further, we are excited for our benchmark to enable future work on prompt design for (one-shot) LLM-based anonymizers, or to develop more lightweight alternatives to models like GPT-4.1, potentially through targeted finetuning – while balancing the risk of *overfitting* (Section 5). Lastly, future work could explore more advanced and possibly domain-specific utility metrics to better navigate the trade-offs, including metrics that measure semantic meaning or specifically target the quality of LLMs post-trained on anonymized chat interactions.
>
> We have now added these points to Appendix K and refer to them in Section 6.
>
> > Matching based on Jaro-Winkler for free-text attributes is an improvement over direct matching. Did you manually verify that this threshold holds here? From personal human post-labeling efforts, this can still be partly off.
>
> We thank the reviewer for this question. We agree that a single global Jaro-Winkler threshold would likely be unreliable, especially for short or closely related category labels. This is why we manually inspected the Jaro-Winkler scores for each attribute on a sample of model outputs and then set an attribute-specific threshold (and simple rules) rather than using one universal cutoff. For example, for “education attainment” we explicitly require the grade level to match exactly and therefore treat “Grade 8” as incorrect when the ground truth is “Grade 9”, even though their Jaro-Winkler similarity is high. More generally, in cases where the string similarity is high but the semantic value differs (e.g., adjacent categories), we adopt a conservative policy and label the prediction as incorrect.
>
> We further elaborate on this in Appendix F and have now updated all results in Table 1 according to this more conservative threshold. We find that the main conclusions remain the same.
>
> **We hope these revisions address your concerns. Please let us know if there are any remaining issues, suggestions, or questions that -- if clarified or addressed -- would contribute to raising the overall score.**

---

> ### Author Response · Authors · 2025-11-21
> **References**
>
> We use the same 6 references as mentioned by the reviewer, and augment the references with:
>
> [7] White, C., Dooley, S., et al. (2025). LiveBench: A challenging, contamination-limited LLM benchmark. (ICLR 2025).
>
> [8] https://openai.com/en-GB/policies/how-your-data-is-used-to-improve-model-performance/
>
> [9] https://techcrunch.com/2025/08/28/anthropic-users-face-a-new-choice-opt-out-or-share-your-data-for-ai-training/
>
> [10] Jennifer King, Kevin Klyman, Emily Capstick, Tiffany Saade, and Victoria Hsieh. User privacy
> and large language models: An analysis of frontier developers’ privacy policies.
>
> [11] https://www.reuters.com/business/media-telecom/openai-fights-order-turn-over-millions-chatgpt-conversations-2025-11-12/

---

> > ### Comment · Reviewer_uffD · 2025-11-26
> > **Thank you**
> >
> > I thank the authors for their extensive answer. I will try to reply roughly point by point:
> >
> > Overall, I agree that cases containing similar information to those in RAT-Bench definitely can occur in practice, as stated in my initial review. At the same time, it is important that the dataset is contextualized properly when it does not reflect the statistics encountered in the real world (this now refers to text-level statistics and not attribute-level statistics, which are matched to the US census). I will expand on this at the end.
> >
> > With respect to the strength of the adversary, I thank the authors for running additional experiments. I found the results interesting, as prior work [1,2,3] has shown that stronger adversaries can make stronger inferences about real texts. Do you think the observed equal performance across models is primarily based on the more explicit nature of RAT-Bench, making Llama-3.1-8B sufficient as adversary?
> >
> > I thank the authors for including some experiments on multi-step anonymization. It is also a good point that accurately estimating which attributes to target here is hard (depending on the use case). Overall, the experimental evaluation here is now satisfactory in my eyes.
> >
> > I also like the inclusion of Appendix I and Table 6. I think such efforts can actually benefit the community.
> >
> > The novelty aspect I am only partly convinced by:
> > You are generating a new dataset that is (by definition) novel, which allows for slightly more control over these attributes.
> > You are applying a (in this specific context) novel metric for re-identification.
> > At the same time
> > The datapoint quality, while grounded in the US Census attribute-wise, is not properly controlled for during generation or post-hoc.
> > Could I not post-hoc sample existing datasets/pipelines (WildChat/SynthPAI) to achieve similar measurements (including indirect attributes) but with more realistic underlying data? For me, too much of the statement "re-identification risk w.r.t. US-Census" relies on the unverified and implicit assumption that the created data is of realistic quality.
> >
> > To be clear, I am not arguing that these types of re-identification risks do not arise in practice (they certainly do), and the overall line of work here has value for improving text anonymization. However, without stronger evidence that the synthetic data meaningfully reflects the distributional properties of real-world texts, the conclusions one can draw from the presented results remain limited. Prior work strengthened this connection by validating against real-world data in the respective domains (or building on real data) - leaving the current version of RAT-Bench feeling a bit subpar. At the same time, the current differentiation between using re-identification risk as in Rocher or reporting the number of indirect identifiers (as has been mentioned in other reviews) seems like a straightforward application of existing work.
> >
> > Based on this, I will keep my score for now and will actively follow the other reviews. I'm open to being convinced by the other reviewers who are in favor of acceptance, but due to the aforementioned reasons, I will not individually favor acceptance.
> >
> >
> > Nit: Line 309 has an unfinished command

---

> > > ### Author Response · Authors · 2025-12-03
> > >
> > > We thank the reviewer for engaging with our rebuttal. We address each of the points raised below.
> > >
> > > >  Do you think the observed equal performance across models is primarily based on the more explicit nature of RAT-Bench, making Llama-3.1-8B sufficient as adversary?
> > >
> > > Thank you for this question. We indeed believe that, as the attributes are leaked in a controlled manner (whether it is level 1, 2 or 3), in cases where attributes are not successfully anonymized, a model of the scale of Llama-3.1-8B is already sufficiently capable of inferring them.
> > >
> > > > I thank the authors for including some experiments on multi-step anonymization. It is also a good point that accurately estimating which attributes to target here is hard (depending on the use case). Overall, the experimental evaluation here is now satisfactory in my eyes.
> > >
> > > We thank the reviewer for suggesting this experiment, and believe it improved our contribution.
> > >
> > > > I also like the inclusion of Appendix I and Table 6. I think such efforts can actually benefit the community.
> > >
> > > We thank the reviewer for suggesting further examination of anonymizers’ failure cases and believe including a more thorough analysis improved our contribution.
> > >
> > > > Could I not post-hoc sample existing datasets/pipelines (WildChat/SynthPAI) to achieve similar measurements (including indirect attributes) but with more realistic underlying data? For me, too much of the statement "re-identification risk w.r.t. US-Census" relies on the unverified and implicit assumption that the created data is of realistic quality. … without stronger evidence that the synthetic data meaningfully reflects the distributional properties of real-world texts, the conclusions one can draw from the presented results remain limited.
> > >
> > > We see two aspects behind the reviewer’s point: (i) the realisticness of the indirect attributes, and (ii) the realisticness of the actual text. We address each separately below.
> > >
> > > **(i) Realisticness of the indirect attributes.** Our work explicitly focuses on computing a *realistic* re-identification risk for a given population. For this, one needs:
> > > Attribute encodings that match the real world.
> > > Representative individual-level data, preserving the full joint distribution of attributes across individuals, not just their marginals.
> > >
> > > Our setup uses attributes from real-world US demographics and therefore incorporates these necessary elements *by design*. Comparatively, the real-world dataset WildChat [9] lacks ground-truth identifiers, so alignment with real-world encodings is not possible. Further, purely synthetic indirect attributes, such as those used in SynthPAI [3], typically do not match real-world attribute encodings, nor do they capture realistic joint distributions.
> > >
> > > Moreover, the correct inference of an attribute is not equally important for each attribute, or individual. For instance, some real-world attribute combinations inherently carry higher re-identification risk (e.g., a 30-year-old personal trainer is far less unique than a 60-year-old one), which relies on faithful modelling of real attribute combinations. We further demonstrate this by computing the Pearson correlation between re-identification risk and the number of correctly inferred indirect identifiers in the table below (mean across anonymization methods for each level). An average correlation of 0.21 confirms that attribute inference accuracy and re-identification risk measure fundamentally different things: re-identification risk depends on the specific values and combinations of attributes, not merely on the binary correctness of inference. Thus, to compute a meaningful re-identification risk, it is essential to incorporate real-world, ground-truth demographic data, rather than rely on real-world text without ground truth (WildChat) or on synthetic data that lacks realistic joint attribute distributions (SynthPAI).
> > >
> > > | Level | Mean Correlation |
> > > |----|----|
> > > | 1 | 0.17 |
> > > | 2 | 0.21 |
> > > | 3 | 0.23 |
> > >
> > > As a framework, we agree that SynthPAI could, in principle, be extended toward our use case, but only if it were (i) seeded with the same real-world indirect identifiers as we use, (ii) include a means of systematically varying feature combinations across different difficulty levels as we do and (iii) compute reidentification risk with our methodology. However, we view these changes as *substantial* and not possible with simple post-hoc sampling.

---

> > > > ### Author Response · Authors · 2025-12-03
> > > >
> > > > **(ii) Realisticness of the text.** We agree with the reviewer and do not argue that our benchmark entries are representative of an *average* real-world medical consultation or chatbot conversation (e.g. consisting of exactly 1 direct and 5 indirect identifiers). Instead, our setup allows us to systematically vary the number, combinations and difficulty level of features represented in our dataset (as shown in Figures 2 and 3), **creating test cases that rigorously evaluate anonymizer performance and measure progress**. This would not be easily achievable with other existing datasets, such as Wildchat and SynthPAI, as they do not provide a comprehensive set of feature combinations distributed across difficulty levels.
> > > >
> > > > Similar to other leading benchmarks [7,8], RAT-Bench creates an artificial environment that prioritises comprehensive and rigorous analysis over realism of test cases. For instance, LiveCodeBench [8] evaluates LLM coding performance on a curated list of coding competition problem sets, which are not realistic representations of the coding problems that LLMs would encounter in practice.
> > > >
> > > > > At the same time, the current differentiation between using re-identification risk as in Rocher or reporting the number of indirect identifiers (as has been mentioned in other reviews) seems like a straightforward application of existing work.
> > > >
> > > > While we of course acknowledge that the mathematical foundation to compute the re-identification risk from a set of indirect identifiers has been established by Rocher et al., we believe that integrating this into an **end-to-end benchmark to evaluate text anonymization tools** is a valuable contribution.
> > > >
> > > > We indeed ensure that, beyond our direct identifiers, our indirect identifiers are grounded in real-world demographics such that, by design, we can link any set of correctly inferred attributes to a realistic re-identification risk following Rocher et al.  This stands in sharp contrast to prior work, which typically evaluates anonymization systems based on their ability to mask specific instances of identifiers [1,2] or the number of attributes that can still be correctly inferred [3,4,5,6], rather than on their impact on real-world re-identification risk.
> > > >
> > > >
> > > > > Nit: Line 309 has an unfinished command
> > > >
> > > > Thank you for noticing, we will fix this error.

---

> > > > > ### Author Response · Authors · 2025-12-03
> > > > > **References**
> > > > >
> > > > > [1] Pilán, Ildikó, et al. "The text anonymization benchmark (tab): A dedicated corpus and evaluation framework for text anonymization." Computational Linguistics 48.4 (2022): 1053-1101.
> > > > >
> > > > > [2] Zhou, Wenxuan, et al. "Universalner: Targeted distillation from large language models for open named entity recognition." arXiv preprint arXiv:2308.03279 (2023).
> > > > >
> > > > > [3] Yukhymenko, Hanna, et al. "A synthetic dataset for personal attribute inference." Advances in Neural Information Processing Systems 37 (2024): 120735-120779.
> > > > >
> > > > > [4] Staab, Robin, et al. "Beyond Memorization: Violating Privacy via Inference with Large Language Models." The Twelfth International Conference on Learning Representations.
> > > > >
> > > > > [5] Staab, Robin, et al. "Language models are advanced anonymizers." The Thirteenth International Conference on Learning Representations. 2025.
> > > > >
> > > > > [6] Liu, Yupei, et al. "Evaluating {LLM-based} Personal Information Extraction and Countermeasures." 34th USENIX Security Symposium (USENIX Security 25). 2025.
> > > > >
> > > > > [7] White, Colin, et al. "LiveBench: A Challenging, Contamination-Limited LLM Benchmark." The Thirteenth International Conference on Learning Representations. 2025.
> > > > >
> > > > > [8] Jain, Naman, et al. "LiveCodeBench: Holistic and Contamination Free Evaluation of Large Language Models for Code." The Thirteenth International Conference on Learning Representations. 2025.
> > > > >
> > > > > [9] Zhao, Wayne Xin, et al. "WildChat: 1M ChatGPT Interaction Logs in the Wild." The Twelfth International Conference on Learning Representations. 2025.

---

### Official Review · Reviewer_RT7r · 2025-11-02

**Soundness:** 3
**Presentation:** 3
**Contribution:** 3
**Rating:** 6
**Confidence:** 3

**Summary:**

This paper discusses the topic of PII removal and identity re-identification in the context of LLMs. In particular, it presents a novel synthetic benchmark aimed at measuring re-identification rates with contemporary anomymization approaches. The paper generates the benchmark synthetically based on entity types sampled from the 5% Public Use Microdata Sample (PMUS). This data is then anonymized using NER- and LLM-based approaches. The benchmark itself then measures re-identification risk against an attack. The authors focus on medical appointment transcripts and AI chatbot interactions as scenarios for the benchmark. Gemini 2.5 Flash is used as the model to generate texts from given attributes. The benchmark is evaluated on several anonymization tools: Azure Language Studio, Presidio, Scrubudub, GliNER, UniNER  (NER-based approaches) as well as Anthropic’s PII purifier prompt used with Gemini 2.5 Flash and Llama-3.1-8B. As the attacker, the authors use Llama-3.1-8B-Instruct in conjunction with an existing attack prompt. The main findings show re-identification rates above 35% across all tools, indicating that the investigated anonymization methods anonymize texts in a far from optimal way. The paper provides additional results on varying difficulty levels for the anonymization as well as varying the number of direct or indirect identifiers. Finally, the paper assesses the utility loss incurred from anonymization and reports that tools that anonymize texts more aggressively yield the highest utility losses, and that LLM-based methods preserve utility better than NER-based ones.

**Strengths:**

The paper provides an interesting benchmark to measure the success rates of anonymization tools in NLP. It shows that recent anonymization tools are still far from perfect, an insight which is likely going to be interesting to the wider research community focussing on privacy in NLP. The paper presents extensive experiments, including algorithm ablations (e.g., with respect to the number of direct and indirect identifiers) and is overall well-written and well-structured.

**Weaknesses:**

* The paper makes repeated use of Llama models for both anonymizing the data, and evaluating the anonymizers on the benchmark. I’m concerned that using the same model family across attack and evaluation can lead to unintended biases. I would encourage the authors to at least discuss this potential issue.
* Related to the above, it would be interesting to see additional models employed for generating benchmark data, to observe how different models affect anonymization performance.
* The paper does not seem to be using any form of human validation of the generated benchmark. It would be interesting to see small-scale human annotation studies verifying that the generated benchmark data satisfies the intended shape of the task (esp. w.r.t. the difficulty levels).
* It would be great if the benchmark could be given a more unique name (for easier references).

**Questions:**

N/A

---

> ### Author Response · Authors · 2025-11-21
> **Response to Reviewer RT7r**
>
> We thank the reviewer for their time spent on the paper and their thoughtful feedback. Below, we address each of the points raised.
>
> > The paper makes repeated use of Llama models for both anonymizing the data, and evaluating the anonymizers on the benchmark. I’m concerned that using the same model family across attack and evaluation can lead to unintended biases.
>
> We thank the reviewer for this suggestion, and acknowledge there could be a bias in using the same model family for both the anonymizer and attacker. To address this, we have instantiated the attacker with GPT-4.1. We select GPT-4.1 as we found GPT-5 (at the time of running this experiment, the most recent and strongest GPT model) often refuses to infer attributes, regardless of whether they are generally considered sensitive (e.g., SSN, credit card number) or not (e.g., occupation). For computational reasons, we evaluate it on a subset of anonymizers, namely Azure, GliNER, and Llama with the Anthropic prompt.
>
>
> We find that the stronger model does not meaningfully impact the resulting re-identification risk, indicating that, for the purposes of this benchmark, Llama-3.1-8B is a sufficient attacker. The full results are included in Appendix L.1 of the updated paper.
>
> > It would be interesting to see additional models employed for generating benchmark data, to observe how different models affect anonymization performance.
>
> We agree that using additional models to generate benchmark entries (currently all generated using Gemini-Flash-2.5 as a cost-efficient, capable model) would further substantiate our findings. As such, we have generated additional benchmark data using the state-of-the-art GPT-5 for the Medical conversation scenario and all levels of difficulty. We then compare the performance of three anonymizers (Azure, GliNER, and Llama with the Anthropic prompt) on the GPT-5 based benchmark data to the original Gemini based benchmark. We include these new results in Appendix B in our updated paper and the samples of the benchmark data we generated using GPT-5 in Appendix D.2
>
>
> Interestingly, we find that there is little difference in re-identification success rate when we use GPT-5 generated benchmark data versus Gemini generated benchmark data for all three anonymizers. This suggests that producing synthetic data with controlled mention of identifying attributes, i.e., following the prompt from Algorithm 4, is a task for which Gemini is already sufficiently capable, and that the additional capabilities of a stronger model do not lead to meaningfully different re-identification risk. We agree that a more detailed study of the generated text and the failure cases it reveals could be a very interesting next step, especially since stronger models may produce more realistic or creative failures. We leave this for future work.
>
> > It would be interesting to see small-scale human annotation studies verifying that the generated benchmark data satisfies the intended shape of the task (esp. w.r.t. the difficulty levels).
>
> We thank the reviewer for this great suggestion. We have conducted a small-scale human validation of our difficulty labels. For 180 randomly sampled benchmark entries, one author (blind to the original labels) read the text and, for a randomly chosen target attribute in that text, assigned a difficulty level based solely on the text. We then compared these annotations to the benchmark’s difficulty labels and found agreement in 94% of cases (Easy attributes: 100%, medium attributes: 95%, hard attributes: 89%). We report the details of this validation, together with the full results, in Appendix E.2.
> > It would be great if the benchmark could be given a more unique name (for easier references).
>
> Ah, we agree with the reviewer. We have now dubbed the benchmark RAT-Bench (Re-identification risk in Anonymized Text Benchmark).
>
> **We hope these revisions address your concerns. Please let us know if there are any remaining issues, suggestions, or questions that -- if clarified or addressed -- would contribute to raising the overall score.**

---

### Meta-Review · Area_Chair_6RHL · 2025-12-29

**Summary:**

The paper proposes a synthetic benchmark for measuring the effectiveness of text anonymization tools at preventing re-identification.  It tackles  an important problem. Offering a benchmark including different levels of difficulty and indirect identifiers to the community can help in the design of anonymization methods.

One important contribution is the methodology to build the synthetic dataset with controlled levels of difficulty, direct and indirect identifiers, following the distribution in the 5% Public Use Microdata Sample (PUMS) (US-
CensusBureau). Another positive aspect is to use the framework from Rocher et al. (2019) to compute the reidentification risk. The paper has also been improved thanks to the reviewers feedback (improve the strength and the diversity of the attackers and anonymizers).

But the paper has also several weaknesses.

The novelty is somehow limited because alternative datasets exist with similar objectives. SynthPAI is focused on attribute inference but not the reidentification risk. However, this can be computed as a postprocessing using the same metric from Rocher et al.

It is dangerous to take such comparisons on this dataset as a
definitive fact and the proposition should be rephrased as a tool to
compare anonymizers in a *specific* setting. Privacy depends on
background knowledge and auxiliary data. Using US demographics is a
way to take one specific background knowledge in consideration.  But
the objective to compute a reidentification risk is not attained. Sentences like "Even when direct identifiers are removed, individuals may still
be re-identified through indirect ones, potentially combined with auxiliary information (Xin et al., 2025). Our benchmark addresses this by directly measuring re-identification risk through what the best attacker can infer,..." is clearly exaggerated.  In particular with the medical application, the demographics of the data set when the data comes from a single department, hospital,...  is not the one of the US.

The conclusion summarized in the abstract that off‑the‑shelf methods are unsafe and that anonmyzation can destroy privacy are not new (But it is worth recalling).

Privacy can only be evaluated in a trade‑off with utility. The authors write "We do believe that choosing the right anonymizer for a given use-case requires balancing trade-offs in privacy, utility, and computational cost." But utility is only measured in BLEU. Hence anonymization methods based on generative models are usually associated with a very bad utility. Utility is mostly task dependent. Eg "My phone number is 312, then 480, then 3820", can be removed in some medical setting without affecting utility.

**Reviewer Concerns:**

RT7r: Concerns have been taken into account.
uffD: Concerns have been partially taken into account. But the main concerns are still present
bkRV: Concerns have been taken into account.
8xbg remains critical even after the discussions

**Reviewer Scores:**

RT7r: keep his score which was already 6
uffD won't change
bkRV: has increased his score.
8xbg won't change

Hence, we will have 6/8/4/2...

---

### Decision · Program_Chairs · 2026-01-26

Reject